# An Upper Bound on the Error Induced by Saddlepoint Approximations—Applications to Information Theory [note 1]

**DOI:** 10.3390/e22060690

**Published:** 2020-06-20

**Authors:** Dadja Anade, Jean-Marie Gorce, Philippe Mary, Samir M. Perlaza

**Affiliations:** 1Laboratoire CITI, a Joint Laboratory between INRIA, the Université de Lyon and the Institut National de Sciences Appliquées (INSA) de Lyon. 6 Av. des Arts, 69621 Villeurbanne, France; dadja.anade-akpo@inria.fr (D.A.); jean-marie.gorce@insa-lyon.fr (J.-M.G.); 2IETR and the Institut National de Sciences Appliquées (INSA) de Rennes, 20 Avenue des Buttes de Coësmes, CS 70839, 35708 Rennes, France; philippe.mary@insa-rennes.fr; 3INRIA, Centre de Recherche de Sophia Antipolis—Méditerranée, 2004 Route des Lucioles, 06902 Sophia Antipolis, France; 4Princeton University, Electrical Engineering Department, Princeton, NJ 08544, USA

**Keywords:** sums of independent and identically random variables, saddlepoint approximation, memoryless channels

## Abstract

This paper introduces an upper bound on the absolute difference between: (a) the cumulative distribution function (CDF) of the sum of a finite number of independent and identically distributed random variables with finite absolute third moment; and (b) a saddlepoint approximation of such CDF. This upper bound, which is particularly precise in the regime of large deviations, is used to study the dependence testing (DT) bound and the meta converse (MC) bound on the decoding error probability (DEP) in point-to-point memoryless channels. Often, these bounds cannot be analytically calculated and thus lower and upper bounds become particularly useful. Within this context, the main results include, respectively, new upper and lower bounds on the DT and MC bounds. A numerical experimentation of these bounds is presented in the case of the binary symmetric channel, the additive white Gaussian noise channel, and the additive symmetric α-stable noise channel.

## 1. Introduction

This paper focuses on approximating the cumulative distribution function (CDF) of sums of a finite number of real-valued independent and identically distributed (i.i.d.) random variables with finite absolute third moment. More specifically, let $Y_{1}$, $Y_{2}$, ⋯, $Y_{n}$, with *n* an integer and 2⩽n<∞, be real-valued random variables with probability distribution $P_{Y}$. Denote by $F_{Y}$ the CDF associated with $P_{Y}$, and, if it exists, denote by $f_{Y}$ the corresponding probability density function (PDF). Let also
(1)$$X_{n}=\sum_{t=1}^{n}Y_{t}$$
be a random variable with distribution $P_{X_{n}}$. Denote by $F_{X_{n}}$ the CDF and if it exists, denote by $f_{X_{n}}$ the PDF associated with $P_{X_{n}}$. The objective is to provide a positive function that approximates $F_{X_{n}}$ and an upper bound on the resulting approximation error. In the following, a positive function $g:R\to R_{+}$ is said to approximate $F_{X_{n}}$ with an *approximation error* that is upper bounded by a function $\epsilon :R\to R_{+}$, if, for all x∈R,
(2)$$|F_{X_{n}}\left(x\right)-g\left(x\right)|⩽\epsilon \left(x\right).$$

The case in which $Y_{1}$, $Y_{2}$, ⋯, $Y_{n}$ in (Equation 1) are stable random variables with $F_{Y}$ analytically expressible is trivial. This is essentially because the sum $X_{n}$ follows the same distribution of a random variable $a_{n}Y+b_{n}$, where $\left(a_{n},b_{n}\right)\in R^{2}$ and *Y* is a random variable whose CDF is $F_{Y}$. Examples of this case are random variables following the Gaussian, Cauchy, or Levy distributions [1].

In general, the problem of calculating the CDF of $X_{n}$ boils down to calculating n−1 convolutions. More specifically, it holds that
(3)$$f_{X_{n}}\left(x\right)=\int_{-\infty }^{\infty }f_{X_{n-1}}\left(x-t\right)f_{Y}\left(t\right)dt,$$
where $f_{X_{1}}=f_{Y}$. Even for discrete random variables and small values of *n*, the integral in (Equation 3) often requires excessive computation resources [2].

When the PDF of the random variable $X_{n}$ cannot be conveniently obtained but only the *r* first moments are known, with r∈N, an approximation of the PDF can be obtained by using an Edgeworth expansion. Nonetheless, the resulting relative error in the large deviation regime makes these approximations inaccurate [3].

When the cumulant generating function (CGF) associated with $F_{Y}$, denoted by $K_{Y}:R\to R$, is known, the PDF $f_{X_{n}}$ can be obtained via the Laplace inversion lemma [2]. That is, given two reals $\alpha_{-}<0$ and $\alpha_{+}>0$, if $K_{Y}$ is analytic for all $z\in \{a\;+ib\in C:\left(a,b\right)\in R^{2}\;\mathrm{and}\;\alpha_{-}⩽a\;⩽\alpha_{+}\}\subset C$, then,
(4)$$f_{X_{n}}\left(x\right)=\frac{1}{2\pi i}\int_{\gamma -i\infty }^{\gamma +i\infty }\exp\left(nK_{Y}\left(z\right)-zx\right)dz,$$
with $i=\sqrt{-1}$ and $\gamma \in (\alpha_{-},\alpha_{+})$. Note that the domain of $K_{Y}$ in (Equation 4) has been extended to the complex plane and thus it is often referred to as the complex CGF. With an abuse of notation, both the CGF and the complex CGF are identically denoted.

In the case in which *n* is sufficiently large, an approximation to the Bromwich integral in (Equation 4) can be obtained by choosing the contour to include the unique saddlepoint of the integrand as suggested in [4]. The intuition behind this lies on the following observations:*(i)* the saddlepoint, denoted by $z_{0}$, is unique, real and $z_{0}\in \left(\alpha_{-},\alpha_{+}\right)$;*(ii)* within a neighborhood around the saddlepoint of the form $\left|z-z_{0}\right|<\epsilon$, with z∈C and ϵ>0 sufficiently small, ImnKY(z)−zx=0 and RenKY(z)−zx can be assumed constant; and*(iii)* outside such neighborhood, the integrand is negligible.

From (i), it follows that the derivative of $nK_{Y}\left(t\right)-tx$ with respect to *t*, with t∈R, is equal to zero when it is evaluated at the saddlepoint $z_{0}$. More specifically, for all t∈R,
(5)$$\frac{d}{dt}K_{Y}\left(t\right)=E_{P_{Y}}\left[Y\exp\left(tY-K_{Y}\left(t\right)\right)\right],$$
and thus
(6)$$E_{P_{Y}}\left[Y\exp\left(z_{0}Y-K_{Y}\left(z_{0}\right)\right)\right]=\frac{x}{n},$$
which shows the dependence of $z_{0}$ on both *x* and *n*.

A Taylor series expansion of the exponent $nK_{Y}\left(z\right)-zx$ in the neighborhood of $z_{0}$, leads to the following asymptotic expansion in powers of $\frac{1}{n}$ of the Bromwich integral in (Equation 4):(7)$$\begin{array}{ccc} f_{X_{n}}\left(x\right) & = & {\hat{f}}_{X_{n}}\left(x\right)\left(1+\frac{1}{n}\left(\frac{1}{8}\frac{K_{Y}^{\left(4\right)}\left(z_{0}\right)}{{\left(K_{Y}^{\left(2\right)}\left(z_{0}\right)\right)}^{2}}-\frac{5}{24}\frac{{\left(K_{Y}^{\left(3\right)}\left(z_{0}\right)\right)}^{2}}{{\left(K_{Y}^{\left(2\right)}\left(z_{0}\right)\right)}^{3}}\right)+O\left(\frac{1}{n^{2}}\right)\right), \end{array}$$
where ${\hat{f}}_{X_{n}}:R\to R_{+}$ is
(8)$$\begin{array}{ccc} {\hat{f}}_{X_{n}}\left(x\right) & = & \sqrt{\frac{1}{2\pi nK_{Y}^{\left(2\right)}\left(z_{0}\right)}}\exp\left(nK_{Y}\left(z_{0}\right)-z_{0}x\right), \end{array}$$
and for all k∈N and t∈R, the notation $K_{Y}^{\left(k\right)}\left(t\right)$ represents the *k*-th real derivative of the CGF $K_{Y}$ evaluated at *t*. The first two derivatives $K_{Y}^{\left(1\right)}$ and $K_{Y}^{\left(2\right)}$ play a central role, and thus it is worth providing explicit expressions. That is,
(9)$$\begin{array}{ccc} K_{Y}^{\left(1\right)}\left(t\right) & ≜ & E_{P_{Y}}\left[Y\exp\left(tY-K_{Y}\left(t\right)\right)\right],\;\mathrm{and} \end{array}$$
(10)$$\begin{array}{ccc} K_{Y}^{\left(2\right)}\left(t\right) & ≜ & E_{P_{Y}}\left[{\left|Y-K_{Y}^{\left(1\right)}\left(t\right)\right|}^{2}\exp\left(tY-K_{Y}\left(t\right)\right)\right]. \end{array}$$

The function ${\hat{f}}_{X_{n}}$ in (Equation 8) is referred to as the *saddlepoint approximation* of the PDF $f_{X_{n}}$ and was first introduced in [4]. Nonetheless, ${\hat{f}}_{X_{n}}$ is not necessarily a PDF as often its integral on R is not equal to one. A particular exception is observed only in three cases [5]. First, when $f_{Y}$ is the PDF of a Gaussian random variable, the saddlepoint approximation ${\hat{f}}_{X_{n}}$ is identical to $f_{X_{n}}$, for all n>0. Second and third, when $f_{Y}$ is the PDF associated with a Gamma distribution and an inverse normal distribution, respectively, the saddlepoint approximation ${\hat{f}}_{X_{n}}$ is exact up to a normalization constant for all n>0.

An approximation to the CDF $F_{X_{n}}$ can be obtained by integrating the PDF in (Equation 4), cf., [6,7,8]. In particular, the result reported in [6] leads to an asymptotic expansion of the CDF of $X_{n}$, for all x∈R, of the form:(11)$$\begin{array}{ccc} F_{X_{n}}\left(x\right) & = & {\hat{F}}_{X_{n}}\left(x\right)+O\left(\frac{1}{\sqrt{n}}\exp\left(nK_{Y}\left(z_{0}\right)-xz_{0}\right)\right), \end{array}$$
where the function ${\hat{F}}_{X_{n}}:R\to R$ is the *saddlepoint approximation* of $F_{X_{n}}$. That is, for all x∈R,
(12)$$\begin{array}{ccc} {\hat{F}}_{X_{n}}\left(x\right) & = & 1_{\left\{z_{0}>0\right\}}+{\left(-1\right)}^{1_{\left\{z_{0}>0\right\}}}\exp\left(nK_{Y}\left(z_{0}\right)-z_{0}x+\frac{1}{2}z_{0}^{2}nK_{Y}^{\left(2\right)}\left(z_{0}\right)\right)Q\left(|z_{0}|\sqrt{nK_{Y}^{\left(2\right)}\left(z_{0}\right)}\right),\; \end{array}$$
where the function Q:R→[0,1] is the complementary CDF of a Gaussian random variable with zero mean and unit variance. That is, for all t∈R,
(13)$$Q\left(t\right)=\frac{1}{\sqrt{2\pi }}\int_{t}^{\infty }\exp\left(-\frac{x^{2}}{2}\right)dx.$$

Finally, from the central limit theorem [3], for large values of *n* and for all x∈R, a reasonable approximation to $F_{X_{n}}\left(x\right)$ is 1−Q(x). In the following, this approximation is referred to as the *normal approximation* of $F_{X_{n}}$.

### 1.1. Contributions

The main contribution of this work is an upper bound on the error induced by the saddlepoint approximation ${\hat{F}}_{X_{n}}$ in (Equation 12) (Theorem 3 in Section 2.2). This result builds upon two observations. The first observation is that the CDF $F_{X_{n}}$ can be written for all x∈R in the form,
(14)$$\begin{array}{c} F_{\;X_{n}\;}\;\left(\;x\;\right)\;=\;1_{\left\{\;z_{0}⩽0\;\right\}}\;E_{\;P_{\;S_{n}\;}\;}\;\left[\;\exp\;\left(\;nK_{\;Y\;}\left(\;z_{0}\;\right)\;-\;z_{0}S_{n}\;\right)\;1_{\left\{\;S_{n}⩽x\;\right\}}\;\right]\;+\;1_{\left\{\;z_{0}>0\;\right\}}\;\left(\;1-\;E_{P_{S_{n}}}\;\left[\;\exp\;\left(\;nK_{Y}\left(z_{0}\right)-z_{0}S_{n}\;\right)\;1_{\left\{S_{n}>x\right\}}\;\right]\;\right),\; \end{array}$$
where the random variable
(15)$$S_{n}=\sum_{t=1}^{n}Y_{t}^{\left(z_{0}\right)}$$
has a probability distribution denoted by $P_{S_{n}}$, and the random variables $Y_{1}^{\left(z_{0}\right)}$, $Y_{2}^{\left(z_{0}\right)}$, *…*, $Y_{n}^{\left(z_{0}\right)}$ are independent with probability distribution $P_{Y^{\left(z_{0}\right)}}$. The distribution $P_{Y^{\left(z_{0}\right)}}$ is an exponentially tilted distribution [9] with respect to the distribution $P_{Y}$ at the saddlepoint $z_{0}$. More specifically, the Radon–Nikodym derivative of the distribution $P_{Y^{\left(z_{0}\right)}}$ with respect to the distribution $P_{Y}$ satisfies for all $y\in \mathrm{supp}P_{Y}$,
(16)$$\begin{array}{c} \frac{dP_{Y^{\left(z_{0}\right)}}}{dP_{Y}}\left(y\right)=\exp\left(-\left(K_{Y}\left(z_{0}\right)-z_{0}y\right)\right). \end{array}$$

The second observation is that the saddlepoint approximation ${\hat{F}}_{X_{n}}$ in (Equation 12) can be written for all x∈R in the form,
(17)$$\begin{array}{c} {\hat{F}}_{\;X_{n}\;}\;\left(\;x\;\right)\;=\;1_{\left\{\;z_{0}⩽0\;\right\}}\;E_{\;P_{\;Z_{n}\;}\;}\;\left[\;\exp\;\left(\;nK_{\;Y\;}\left(\;z_{0}\;\right)\;-\;z_{0}Z_{n}\;\right)\;1_{\left\{\;Z_{n}⩽x\;\right\}}\;\right]\;+\;1_{\left\{\;z_{0}>0\;\right\}}\;\left(\;1-\;E_{P_{Z_{n}}}\;\left[\;\exp\;\left(\;nK_{Y}\left(z_{0}\right)-z_{0}Z_{n}\;\right)\;1_{\left\{Z_{n}>x\right\}}\;\right]\;\right)\;,\; \end{array}$$
where $Z_{n}$ is a Gaussian random variable with mean *x*, variance $nK_{Y}^{\left(2\right)}\left(z_{0}\right)$, and probability distribution $P_{Z_{n}}$. Note that the means of the random variable $S_{n}$ in (Equation 14) and $Z_{n}$ in (Equation 17) are equal to $nK_{Y}^{\left(1\right)}\left(z_{0}\right)$, whereas their variances are equal to $nK_{Y}^{\left(2\right)}\left(z_{0}\right)$. Note also that, from (Equation 6), it holds that $x=nK_{Y}^{\left(1\right)}\left(z_{0}\right)$.

Using these observations, it holds that the absolute difference between $F_{X_{n}}$ in (Equation 14) and ${\hat{F}}_{X_{n}}$ in (Equation 17) satisfies for all x∈R,
(18)$$\begin{array}{c} \left|F_{X_{n}}\left(x\right)-{\hat{F}}_{X_{n}}\left(x\right)\right|\\ =1_{\left\{z_{0}⩽0\right\}}\left|E_{P_{S_{n}}}\left[\exp\left(nK_{Y}\left(z_{0}\right)-z_{0}S_{n}\right)1_{\left\{S_{n}⩽x\right\}}\right]-E_{P_{Z_{n}}}\left[\exp\left(nK_{Y}\left(z_{0}\right)-z_{0}Z_{n}\right)1_{\left\{Z_{n}⩽x\right\}}\right]\right|\\ +1_{\left\{z_{0}>0\right\}}\left|E_{P_{S_{n}}}\left[\exp\left(nK_{Y}\left(z_{0}\right)-z_{0}S_{n}\right)1_{\left\{S_{n}>x\right\}}\right]-E_{P_{Z_{n}}}\left[\exp\left(nK_{Y}\left(z_{0}\right)-z_{0}Z_{n}\right)1_{\left\{Z_{n}>x\right\}}\right]\right|.\; \end{array}$$

A step forward (Lemma A1 in Appendix A) is to note that, when *x* is such that $z_{0}⩽0$, then,
(19)$$\begin{array}{c} \left|E_{P_{S_{n}}}\left[\exp\left(nK_{Y}\left(z_{0}\right)-z_{0}S_{n}\right)1_{\left\{S_{n}⩽x\right\}}\right]-E_{P_{Z_{n}}}\left[\exp\left(nK_{Y}\left(z_{0}\right)-z_{0}Z_{n}\right)1_{\left\{Z_{n}⩽x\right\}}\right]\right|\\ ⩽\exp\left(nK_{Y}\left(z_{0}\right)-z_{0}x\right)\min\left\{1,2\underset{a\in R}{\sup}\left|F_{S_{n}}\left(a\right)-F_{Z_{n}}\left(a\right)\right|\right\}, \end{array}$$
and when *x* is such that $z_{0}>0$, it holds that
(20)$$\begin{array}{c} \left|E_{P_{S_{n}}}\left[\exp\left(nK_{Y}\left(z_{0}\right)-z_{0}S_{n}\right)1_{\left\{S_{n}>x\right\}}\right]-E_{P_{Z_{n}}}\left[\exp\left(nK_{Y}\left(z_{0}\right)-z_{0}Z_{n}\right)1_{\left\{Z_{n}>x\right\}}\right]\right|\\ ⩽\exp\left(nK_{Y}\left(z_{0}\right)-z_{0}x\right)\min\left\{1,2\underset{a\in R}{\sup}\left|F_{S_{n}}\left(a\right)-F_{Z_{n}}\left(a\right)\right|\right\}, \end{array}$$
where $F_{S_{n}}$ and $F_{Z_{n}}$ are the CDFs of the random variables $S_{n}$ and $Z_{n}$, respectively. The final result is obtained by observing that $\sup_{a\in R}\left|F_{S_{n}}\left(a\right)-F_{Z_{n}}\left(a\right)\right|$ can be upper bounded using the Berry–Esseen Theorem (Theorem 1 in Section 2.1). This is essentially due to the fact that the random variable $S_{n}$ is the sum of *n* independent random variables, i.e., (Equation 15), and $Z_{n}$ is a Gaussian random variable, and both $S_{n}$ and $Z_{n}$ possess identical means and variances. Thus, the main result (Theorem 3 in Section 2.2) is that, for all x∈R,
(21)$$\begin{array}{c} \left|F_{X_{n}}\left(x\right)-{\hat{F}}_{X_{n}}\left(x\right)\right|⩽\frac{2\xi_{Y}\left(z_{0}\right)}{\sqrt{n}}\exp\left(nK_{Y}\left(z_{0}\right)-z_{0}\;x\right), \end{array}$$
where
(22)$$\begin{array}{c} \xi_{Y}\left(z_{0}\right)=c_{1}\left(\frac{E_{P_{Y}}\left[{\left|Y-K_{Y}^{\left(1\right)}\left(z_{0}\right)\right|}^{3}\exp\left(z_{0}Y-K_{Y}\left(z_{0}\right)\right)\right]}{{\left(K_{Y}^{\left(2\right)}\left(z_{0}\right)\right)}^{3/2}}+c_{2}\right), \end{array}$$
with
(23a)$$\begin{array}{c} c_{1}≜0.33554,\mathrm{and} \end{array}$$
(23b)$$\begin{array}{c} c_{2}≜0.415. \end{array}$$

Finally, note that (Equation 21) holds for any finite value of *n* and admits the asymptotic scaling law with respect to *n* suggested in (Equation 11).

### 1.2. Applications

In the realm of information theory, the normal approximation has played a central role in the calculation of bounds on the minimum decoding error probability (DEP) in point-to-point memoryless channels, cf., [10,11]. Thanks to the normal approximation, simple approximations for the dependence testing (DT) bound, the random coding union bound (RCU) bound, and the meta converse (MC) bound have been obtained in [10,12]. The success of these approximations stems from the fact that they are easy to calculate. Nonetheless, easy computation comes at the expense of loose upper and lower bounds and thus uncontrolled approximation errors.

On the other hand, saddlepoint techniques have been extensively used to approximate existing lower and upper bounds on the minimum DEP. See, for instance, [13,14] in the case of the RCU bound and the MC bound. Nonetheless, the errors induced by saddlepoint approximations are often neglected due to the fact that calculating them involves a large number of optimizations and numerical integrations. Currently, the validation of saddlepoint approximations is carried through Monte Carlo simulations. Within this context, the main objectives of this paper are twofold: (a) to analytically assess the tightness of the approximation of DT and MC bounds based on the saddlepoint approximation of the CDFs of sums of i.i.d. random variables; (b) to provide new lower and upper bounds on the minimum DEP by providing a lower bound on the MC bound and an upper bound on the DT bound. Numerical experimentation of these bounds is presented for the binary symmetric channel (BSC), the additive white Gaussian noise (AWGN) channel, and the additive symmetric α-stable noise (SαS) channel, where the new bounds are tight and obtained at low computational cost.

## 2. Sums of Independent and Identically Distributed Random Variables

In this section, upper bounds on the absolute error of approximating $F_{X_{n}}$ by the *normal approximation* and the *saddlepoint approximation* are presented.

### 2.1. Error Induced by the Normal Approximation

Given a random variable *Y*, let the function $\xi_{Y}:R\overset{}{\to }R$ be for all t∈R :(24)$$\begin{array}{ccc} \xi_{Y}\left(t\right) & ≜ & c_{1}\left(\frac{E_{P_{Y}}\left[{\left|Y-K_{Y}^{\left(1\right)}\left(t\right)\right|}^{3}\exp\left(tY-K_{Y}\left(t\right)\right)\right]}{{\left(K_{Y}^{\left(2\right)}\left(t\right)\right)}^{3/2}}+c_{2}\right), \end{array}$$
where $c_{1}$ and $c_{2}$ are defined in (23).

The following theorem, known as the Berry–Esseen theorem [3], introduces an upper bound on the approximation error induced by the normal approximation.

**Theorem** **1**(Berry–Esseen [15]). *Let $Y_{1}$, $Y_{2}$, …, $Y_{n}$ be i.i.d random variables with probability distribution $P_{Y}$. Let also $Z_{n}$ be a Gaussian random variable with mean $n\;K_{Y}^{\left(1\right)}\left(0\right)$, variance $n\;K_{Y}^{\left(2\right)}\left(0\right)$, and CDF denoted by $F_{Z_{n}}$. Then, the CDF of the random variable $X_{n}$=$Y_{1}$+$Y_{2}$+…+$Y_{n}$, denoted by $F_{X_{n}}$, satisfies*
(25)$$\begin{array}{c} \underset{a\in R}{\sup}\left|F_{X_{n}}\left(a\right)-F_{Z_{n}}\left(a\right)\right|⩽\min\left\{1,\frac{\xi_{Y}\left(0\right)}{\sqrt{n}}\right\}, \end{array}$$
*where the functions $K_{Y}^{\left(1\right)}$, $K_{Y}^{\left(2\right)}$ and $\xi_{Y}$ are defined in (Equation 9), (10), and (Equation 24).*


An immediate result from Theorem 1 gives the following upper and lower bounds on $F_{X_{n}}\left(a\right)$, for all a∈R,
(26)$$\begin{array}{c} F_{X_{n}}\left(a\right)⩽F_{Z_{n}}\left(a\right)+\min\left\{1,\frac{\xi_{Y}\left(0\right)}{\sqrt{n}}\right\}≜\bar{\Sigma }\left(a,n\right),\mathrm{and} \end{array}$$
(27)$$\begin{array}{c} F_{X_{n}}\left(a\right)⩾F_{Z_{n}}\left(a\right)-\min\left\{1,\frac{\xi_{Y}\left(0\right)}{\sqrt{n}}\right\}≜\underline{\Sigma }\left(a,n\right). \end{array}$$

The main drawback of Theorem 1 is that the upper bound on the approximation error does not depend on the exact value of *a*. More importantly, for some values of *a* and *n*, the upper bound on the approximation error might be particularly big, which leads to irrelevant results.

### 2.2. Error Induced by the Saddlepoint Approximation

The following theorem introduces an upper bound on the approximation error induced by approximating the CDF $F_{X_{n}}$ of $X_{n}$ in (Equation 1) by the function $\eta_{Y}:$$R^{2}$×N→R defined such that for all (θ, *a*, n)∈$R^{2}$×N,
(28)$$\begin{array}{c} \eta_{Y}\left(\;\theta ,\;a,\;n\;\right)\;\\ ≜1_{\left\{\;\theta >0\;\right\}}\;+\;{\left(\;-\;1\right)}^{1_{\left\{\;\theta >0\;\right\}}}\;\exp\;\left(\;\frac{1}{2}n\theta^{2}K_{Y}^{\left(2\right)}\left(\;\theta \;\right)\;+\;nK_{Y}\left(\theta \right)\;-\;n\theta K_{Y}^{\left(1\right)}\left(\;\theta \;\right)\;\right)Q\;\left(\;{\left(\;-\;1\;\right)}^{1_{\left\{\;\theta ⩽0\;\right\}}}\frac{a\;+\;n\theta K_{Y}^{\left(2\right)}\left(\;\theta \;\right)\;-\;nK_{Y}^{\left(1\right)}\left(\theta \right)}{\sqrt{nK_{Y}^{\left(2\right)}\left(\theta \right)}}\;\right),\; \end{array}$$
where the function Q:R→[0,1] is the complementary CDF of the standard Gaussian distribution defined in (Equation 13). Note that $\eta_{Y}\left(\theta ,n,a\right)$ is identical to ${\hat{F}}_{X_{n}}\left(a\right)$, when θ is chosen to satisfy the saddlepoint $K_{Y}^{\left(1\right)}\left(\theta \right)=\frac{a}{n}$. Note also that $\eta_{Y}\left(0,n,a\right)$ is the CDF of a Gaussian random variable with mean $nK_{Y}^{\left(1\right)}\left(0\right)$ and variance $nK_{Y}^{\left(2\right)}\left(0\right)$, which are the mean and the variance of $X_{n}$ in (Equation 1), respectively.

**Theorem** **2.**
*Let $Y_{1}$, $Y_{2}$, …, $Y_{n}$ be i.i.d. random variables with probability distribution $P_{Y}$ and CGF $K_{Y}$. Let also $F_{X_{n}}$ be the CDF of the random variable $X_{n}$=$Y_{1}$+$Y_{2}$+…+$Y_{n}$. Hence, for all a∈R and for all $\theta \in \Theta_{Y}$, it holds that*
(29)$$\begin{array}{c} \left|F_{X_{n}}\left(a\right)-\eta_{Y}\left(\theta ,a,n\right)\right|⩽\exp\left(nK_{Y}\left(\theta \right)-\theta \;a\;\right)\min\left\{1,\frac{2\;\xi_{Y}\left(\theta \right)}{\sqrt{n}}\right\}, \end{array}$$
*where*
(30)$$\begin{array}{c} \Theta_{Y}≜\left\{t\in R:K_{Y}\left(t\right)<\infty \right\}; \end{array}$$

*and the functions $\xi_{Y}$ and $\eta_{Y}$ are defined in (Equation 24) and (Equation 28), respectively.*


**Proof.** The proof of Theorem 2 is presented in Appendix A. ☐

This result leads to the following upper and lower bounds on $F_{X_{n}}\left(a\right)$, for all a∈R,
(31)$$\begin{array}{ccc} F_{X_{n}}\left(a\right) & ⩽ & \eta_{Y}\left(\theta ,a,n\right)+\exp\left(nK_{Y}\left(\theta \right)-\theta \;a\;\right)\min\left\{1,\frac{2\;\xi_{Y}\left(\theta \right)}{\sqrt{n}}\right\},\mathrm{and} \end{array}$$
(32)$$\begin{array}{ccc} F_{X_{n}}\left(a\right) & ⩾ & \eta_{Y}\left(\theta ,a,n\right)-\exp\left(nK_{Y}\left(\theta \right)-\theta \;a\;\right)\min\left\{1,\frac{2\;\xi_{Y}\left(\theta \right)}{\sqrt{n}}\right\}, \end{array}$$
with θ∈$\Theta_{Y}$.

The advantages of approximating $F_{X_{n}}$ by using Theorem 2 instead of Theorem 1 are twofold. First, both the approximation $\eta_{Y}$ and the corresponding approximation error depend on the exact value of *a*. In particular, the approximation can be optimized for each value of *a* via the parameter θ. Second, the parameter θ in (Equation 29) can be optimized to improve either the upper bound in (Equation 31) or the lower bound in (32) for some a∈R. Nonetheless, such optimizations are not necessarily simple.

An alternative to the optimization on θ in (Equation 31) and (32) is to choose θ such that it minimizes $nK_{Y}\left(\theta \right)-\theta \;a$. This follows the intuition that, for some values of *a* and *n*, the term $\exp(nK_{Y}\left(\theta \right)-\theta \;a)$ is the one that influences the most the value of the right-hand side of (Equation 29). To build upon this idea, consider the following lemma.

**Lemma** **1.**
*Consider a random variable Y with probability distribution $P_{Y}$ and CGF $K_{Y}$. Given n∈N, let the function h:R→R be defined for all a∈R satisfying $\frac{a}{n}\in \mathrm{int}C_{Y}$, with $\mathrm{int}C_{Y}$ denoting the interior of the convex hull of $\mathrm{supp}\;P_{X_{n}}$, as follows:*
(33)$$\begin{array}{c} h\left(a\right)=\underset{\theta \in \Theta_{Y}}{\inf}nK_{Y}\left(\theta \right)-\theta \;a, \end{array}$$
*where $\Theta_{Y}$ is defined in (Equation 30). Then, the function h is concave and for all a∈R,*
(34)$$\begin{array}{c} h\left(a\right)⩽h(nE_{P_{Y}}\left[Y\right])=0. \end{array}$$
*Furthermore,*
(35)$$\begin{array}{c} h\left(a\right)=nK_{Y}\left(\theta^{⋆}\right)-\theta^{⋆}\;a, \end{array}$$
*where $\theta^{⋆}$ is the unique solution in θ to*
(36)$$\begin{array}{c} nK_{Y}^{\left(1\right)}\left(\theta \right)=a, \end{array}$$

*with $K_{Y}^{\left(1\right)}$ is defined in (Equation 9).*


**Proof.** The proof of Lemma 1 is presented in Appendix B. ☐

Given (a,n)∈R×N, the value of h(a) in (Equation 33) is the argument that minimizes the exponential term in (Equation 29). An interesting observation from Lemma 1 is that the maximum of *h* is zero, and it is reached when $a=nE_{P_{Y}}\left[Y\right]=E_{P_{X_{n}}}\left[X_{n}\right]$. In this case, $\theta^{⋆}=0$, and thus, from (Equation 31) and (32), it holds that
$$\begin{array}{ccc} F_{X_{n}}\left(a\right) & ⩽ & \eta_{Y}\left(0,a,n\right)+\min\left\{1,\frac{2\;\xi_{Y}\left(0\right)}{\sqrt{n}}\right\} \end{array}$$
(37)$$\begin{array}{ccc} & = & F_{Z_{n}}\left(a\right)+\min\left\{1,\frac{2\;\xi_{Y}\left(0\right)}{\sqrt{n}}\right\},\mathrm{and}\\ F_{X_{n}}\left(a\right) & ⩾ & \eta_{Y}\left(0,a,n\right)-\min\left\{1,\frac{2\;\xi_{Y}\left(0\right)}{\sqrt{n}}\right\} \end{array}$$
(38)$$\begin{array}{ccc} & = & F_{Z_{n}}\left(a\right)-\min\left\{1,\frac{2\;\xi_{Y}\left(0\right)}{\sqrt{n}}\right\}, \end{array}$$
where $F_{Z_{n}}$ is the CDF defined in Theorem 1. Hence, the upper bound in (Equation 37) and the lower bound in (38) obtained from Theorem 2 are worse than those in (Equation 26) and (27) obtained from Theorem 1. In a nutshell, for values of *a* around the vicinity of $nE_{P_{Y}}\left[Y\right]=E_{P_{X_{n}}}\left[X_{n}\right]$, it is more interesting to use Theorem 1 instead of Theorem 2.

Alternatively, given that *h* is non-positive and concave, when $\left|a\;-nE_{P_{Y}}\left[Y\right]\right|$=|a−$E_{P_{X_{n}}}\left[X_{n}\right]|$>γ, with γ sufficiently large, it follows that
(39)$$\begin{array}{c} \exp\left(nK_{Y}\left(\theta^{⋆}\right)-\theta^{⋆}\;a\;\right)<\min\left\{1,\frac{\xi_{Y}\left(0\right)}{\sqrt{n}}\right\}, \end{array}$$
with $\theta^{⋆}$ defined in (Equation 36). Hence, in this case, the right-hand side of (Equation 29) is always smaller than the right-hand side of (Equation 25). That is, for such values of *a* and *n*, the upper and lower bounds in (Equation 31) and (32) are better than those in (Equation 26) and (27), respectively. The following theorem leverages this observation.

**Theorem** **3.**
*Let $Y_{1}$, $Y_{2}$, …, $Y_{n}$ be i.i.d. random variables with probability distribution $P_{Y}$ and CGF $K_{Y}$. Let also $F_{X_{n}}$ be the CDF of the random variable $X_{n}=Y_{1}+Y_{2}+\ldots +Y_{n}$. Hence, for all a∈$\mathrm{int}\;C_{X_{n}}$, with $\mathrm{int}\;C_{X_{n}}$ the interior of the convex hull of $\mathrm{supp}P_{X_{n}}$, it holds that*
(40)$$\begin{array}{c} \left|F_{X_{n}}\left(a\right)-{\hat{F}}_{X_{n}}\left(a\right)\right|⩽\exp\left(nK_{Y}\left(\theta^{⋆}\right)-\theta^{⋆}\;a\right)\min\left\{1,\frac{2\;\xi_{Y}\left(\theta^{⋆}\right)}{\sqrt{n}}\right\}, \end{array}$$
*where $\theta^{⋆}$ is defined in (Equation 36), and the functions ${\hat{F}}_{X_{n}}$ and $\xi_{Y}$ are defined in (Equation 12), and (Equation 24), respectively.*


**Proof.** The proof of Theorem 3 is presented in Appendix C. ☐

An immediate result from Theorem 3 gives the following upper and lower bounds on $F_{X}\left(a\right)$, for all a∈R ,
(41)$$\begin{array}{ccc} F_{X_{n}}\left(a\right) & ⩽ & {\hat{F}}_{X_{n}}\left(a\right)+\exp\left(nK_{Y}\left(\theta^{⋆}\right)-\theta^{⋆}\;a\right)\min\left\{1,\frac{2\;\xi_{Y}\left(\theta^{⋆}\right)}{\sqrt{n}}\right\}≜\bar{\Omega }\left(a,n\right),\mathrm{and}\; \end{array}$$
(42)$$\begin{array}{ccc} F_{X_{n}}\left(a\right) & ⩾ & {\hat{F}}_{X_{n}}\left(a\right)-\exp\left(nK_{Y}\left(\theta^{⋆}\right)-\theta^{⋆}\;a\right)\min\left\{1,\frac{2\;\xi_{Y}\left(\theta^{⋆}\right)}{\sqrt{n}}\right\}≜\underline{\Omega }\left(a,n\right). \end{array}$$

The following section presents two examples that highlight the observations mentioned above.

### 2.3. Examples

**Example** **1**(Discrete random variable). *Let the random variables $Y_{1}$, $Y_{2}$, …, $Y_{n}$ in (Equation 1) be i.i.d. Bernoulli random variables with parameter p=0.2 and n=100. In this case, $E_{P_{X_{n}}}\left[X_{n}\right]=nE_{P_{Y}}\left[Y\right]=20$. Figure 1 depicts the CDF $F_{X_{100}}$ of $X_{100}$ in (Equation 1); the normal approximation $F_{Z_{100}}$ in (Equation 25); and the saddlepoint approximation ${\hat{F}}_{X_{100}}$ in (Equation 12). Therein, it is also depicted the upper and lower bounds due to the normal approximation $\bar{\Sigma }$ in (Equation 26) and $\underline{\Sigma }$ in (27), respectively; and the upper and lower bounds due to the saddlepoint approximation $\bar{\Omega }$ in (Equation 41) and $\underline{\Omega }$ in (42), respectively. These functions are plotted as a function of a, with a∈[5,35].*

**Example** **2**(Continuous random variable). *Let the random variables $Y_{1}$, $Y_{2}$, …, $Y_{n}$ in (Equation 1) be i.i.d. chi-squared random variables with parameter k=1 and n=50. In this case, $E_{P_{X_{n}}}\left[X_{n}\right]=nE_{P_{Y}}\left[Y\right]=50$. Figure 2 depicts the CDF $F_{X_{50}}$ of $X_{50}$ in (Equation 1); the normal approximation $F_{Z_{50}}$ in (Equation 25); and the saddlepoint approximation ${\hat{F}}_{X_{50}}$ in (Equation 12). Therein, it is also depicted the upper and lower bounds due to the normal approximation $\bar{\Sigma }$ in (Equation 26) and $\underline{\Sigma }$ in (27), respectively; and the upper and lower bounds due to the saddlepoint approximation $\bar{\Omega }$ in (Equation 41) and $\underline{\Omega }$ in (42), respectively. These functions are plotted as a function of a, with a∈[0,100].*

## 3. Application to Information Theory: Channel Coding

This section focuses on the study of the DEP in point-to-point memoryless channels. The problem is formulated in Section 3.1. The main results presented in this section consist of lower and upper bounds on the DEP. The former, which are obtained building upon the existing DT bound [10], are presented in Section 3.2. The latter, which are obtained from the MC bound [10], are presented in Section 3.3.

### 3.1. System Model

Consider a point-to-point communication in which a transmitter aims at sending information to one receiver through a noisy memoryless channel. Such a channel can be modeled by a random transformation
(43)$$\begin{array}{c} (X^{n},Y^{n},P_{Y|X}), \end{array}$$
where n∈N is the blocklength and X and Y are the channel input and channel output sets. Given the channel inputs x=$(x_{1}$, $x_{2}$, *…*, $x_{n})$∈$X^{n}$, the outputs y=$(y_{1}$, $y_{2}$, *…*, $y_{n})$∈$Y^{n}$ are observed at the receiver with probability
(44)$$\begin{array}{c} P_{Y|X}\left(y|x\right)=\prod_{t=1}^{n}P_{Y|X}\left(y_{t}|x_{t}\right), \end{array}$$
where, for all x∈X, $P_{Y|X=x}\in ▵\left(Y\right)$, with $▵\left(Y\right)$, the set of all possible probability distributions whose support is a subset of Y. The objective of the communication is to transmit a message index *i*, which is a realization of a random variable *W* that is uniformly distributed over the set
(45)$$\begin{array}{c} W≜\{1,2,\ldots ,M\}, \end{array}$$
with 1 <*M*<*∞*. To achieve this objective, the transmitter uses an (n, *M*, λ)-code, where λ∈[0,1].

**Definition** **1**((n, *M*,λ)-code). *Given a tuple (M, n, λ)∈$N^{2}$×[0,1], an (n, M, λ)-code for the random transformation in (Equation 43) is a system*
(46)$$\begin{array}{c} \left\{\left(u(1),D(1)\right),\left(u(2),D(2)\right),\ldots ,\left(u(M),D(M)\right)\right\}, \end{array}$$
*where for all $\left(j,\ell \right)\in W^{2}$, with j≠ℓ:*
(47a)$$\begin{array}{ccc} u(j) & = & \left(u_{1}\left(j\right),u_{2}\left(j\right),\ldots ,u_{n}\left(j\right)\right)\in X^{n}, \end{array}$$
(47b)$$\begin{array}{ccc} D(j)\cap D(\ell ) & = & \emptyset , \end{array}$$
(47c)$$\begin{array}{ccc} \underset{j\in W}{⋃}D\left(j\right) & \subseteq & Y^{n},\mathrm{and} \end{array}$$
(47d)$$\begin{array}{ccc} \frac{1}{M}\sum_{i=1}^{M}E_{P_{Y|X=u(i)}}\left[1_{\left\{Y\notin D(i)\right\}}\right] & ⩽ & \lambda . \end{array}$$

To transmit message index *i*∈W, the transmitter uses the codeword u(i). For all *t*∈{ 1,2,*…*, n}, at channel use *t*, the transmitter inputs the symbol $u_{t}\left(i\right)$ into the channel. Assume that, at the end of channel use *t*, the receiver observes the output $y_{t}$. After *n* channel uses, the receiver uses the vector y=$(y_{1}$,$y_{2}$,*…*, $y_{n})$ and determines that the symbol *j* was transmitted if y∈D(j), with *j*∈W.

Given the (n,*M*,λ)-code described by the system in (Equation 46), the DEP of the message index *i* can be computed as $E_{P_{Y|X=u(i)}}[$$1_{\left\{Y\notin D(i)\right\}}]$. As a consequence, the average DEP is
(48)$$\frac{1}{M}\sum_{i=1}^{M}E_{P_{Y|X=u(i)}}\left[1_{\left\{Y\notin D(i)\right\}}\right].$$

Note that, from (47d), the average DEP of such an (n,M,λ)-code is upper bounded by λ. Given a fixed pair (n,M)∈$N^{2}$, the minimum λ for which an (n,*M*,λ)-code exists is defined hereunder.

**Definition** **2.**
*Given a pair (n,M)∈$N^{2}$, the minimum average DEP for the random transformation in (Equation 43), denoted by $\lambda^{*}\left(n,M\right)$, is given by*
(49)$$\begin{array}{c} \lambda^{*}\left(n,M\right)=\min\left\{\lambda \in [0,1]:\exists (n,M,\lambda )-\mathrm{code}\right\}. \end{array}$$


When λ is chosen accordingly with the reliability constraints, an (n,M,λ)-code is said to transmit at an information rate $R=\frac{\log_{2}\left(M\right)}{n}$ bits per channel use.

The remainder of this section introduces the DT and MC bounds. The DT bound is one of the tightest existing upper bounds on $\lambda^{*}\left(n,M\right)$ in (Equation 49), whereas the MC bound is one of the tightest lower bounds.

### 3.2. Dependence Testing Bound

This section describes an upper bound on $\lambda^{*}\left(n,M\right)$, for a fixed pair $\left(n,M\right)\in N^{2}$. Given a probability distribution $P_{X}\in ▵\left(X^{n}\right)$, let the random variable $\iota \left(X;Y\right)$ satisfy
(50)$$\iota \left(X;Y\right)≜\ln\left(\frac{dP_{XY}}{dP_{X}P_{Y}}\left(X,Y\right)\right),$$
where the function $\frac{dP_{XY}}{dP_{X}P_{Y}}:X^{n}\times Y^{n}\to R$ denotes the Radon–Nikodym derivative of the joint probability measure $P_{XY}$ with respect to the product of probability measures $P_{X}P_{Y}$, with $P_{XY}=P_{X}P_{Y|X}$ and $P_{Y}$ the corresponding marginal. Let the function $T:N^{2}\times ▵\left(X^{n}\right)\to R_{+}$ be for all (n,M)∈$N^{2}$ and for all probability distributions $P_{X}\in ▵\left(X^{n}\right)$,
(51)$$\begin{array}{ccc} T(n,M,P_{X}) & = & E_{P_{X}P_{Y|X}}\left[1_{\left\{\iota \left(X;Y\right)⩽\ln\left(\frac{M-1}{2}\right)\right\}}\right]+\frac{M-1}{2}E_{P_{X}P_{Y}}\left[1_{\left\{\iota \left(X;Y\right)>\ln\left(\frac{M-1}{2}\right)\right\}}\right].\; \end{array}$$

Using this notation, the following lemma states the DT bound.

**Lemma** **2**(Dependence testing bound [10]). *Given a pair (n,M)∈$N^{2}$, the following holds for all $P_{X}\in ▵\left(X^{n}\right)$, with respect to the random transformation in (Equation 43):*
(52)$$\begin{array}{c} \lambda^{*}\left(n,M\right)⩽T\left(n,M,P_{X}\right), \end{array}$$
*with the function T defined in (Equation 51).*


Note that the input probability distribution $P_{X}$ in Lemma 2 can be chosen among all possible probability distributions $P_{X}\in ▵\left(X^{n}\right)$ to minimize the right-hand side of (Equation 52), which improves the bound. Note also that with some loss of optimality, the optimization domain can be restricted to the set of product probability distributions for which for all x∈$X^{n}$,
(53)$$\begin{array}{c} P_{X}\left(x\right)=\prod_{t=1}^{n}P_{X}\left(x_{t}\right), \end{array}$$
with $P_{X}\in ▵\left(X\right)$. Hence, subject to (Equation 44), the random variable ι(X;Y) in (Equation 50) can be written as the sum of i.i.d. random variables, i.e.,
(54)$$\begin{array}{c} \iota \left(X;Y\right)=\sum_{t=1}^{n}\iota \left(X_{t};Y_{t}\right). \end{array}$$

This observation motivates the application of the results of Section 2 to provide upper and lower bounds on the function *T* in (Equation 51), for some given values $\left(n,M\right)\in N^{2}$ and a given distribution $P_{X}\in ▵\left(X^{n}\right)$ for the random transformation in (Equation 43) subject to (Equation 44). These bounds become significantly relevant when the exact value of $T(n,M,P_{X})$ cannot be calculated with respect to the random transformation in (Equation 43). In such a case, providing upper and lower bounds on $T(n,M,P_{X})$ helps in approximating its exact value subject to an error sufficiently small such that the approximation is relevant.

#### 3.2.1. Normal Approximation

This section describes the normal approximation of the function *T* in (Equation 51). That is, the random variable ι(X;Y) is assumed to satisfy (Equation 54) and to follow a Gaussian distribution. More specifically, for all $P_{X}\in ▵\left(X\right)$, let
(55)$$\begin{array}{c} \mu \left(P_{X}\right)≜E_{P_{X}P_{Y|X}}\left[\iota (X;Y)\right], \end{array}$$
(56)$$\begin{array}{c} \sigma \left(P_{X}\right)≜E_{P_{X}P_{Y|X}}\left[{\left(\iota \left(X;Y\right)-\mu \left(P_{X}\right)\right)}^{2}\right],\mathrm{and} \end{array}$$
(57)$$\begin{array}{c} \xi \left(P_{X}\right)≜c_{1}\left(\frac{E_{P_{X}P_{Y|X}}\left[|\iota \left(X;Y\right)-\mu \left(P_{X}\right){|}^{3}\right]}{\sigma {\left(P_{X}\right)}^{\frac{3}{2}}}+c_{2}\right), \end{array}$$
with $c_{1}$ and $c_{2}$ defined in (23), be functions of the input distribution $P_{X}$. In particular, $\mu (P_{X})$ and $\sigma (P_{X})$ are respectively the first moment and the second central moment of the random variables $\iota (X_{1};Y_{1})$, $\iota (X_{2};Y_{2})$*…*$\iota (X_{n};Y_{n})$. Using this notation, consider the functions $D:N^{2}\times ▵\left(X\right)\to R_{+}$ and $N:N^{2}\times ▵\left(X\right)\to R_{+}$ such that for all $\left(n,M\right)\in N^{2}$ and for all $P_{X}\in ▵\left(X\right)$,
(58)$$\begin{array}{c} D\left(n,M,P_{X}\right)=\max\;\left\{0,\alpha \;\left(\;n,M,P_{X}\right)\;-\;\frac{\xi (P_{X})}{\sqrt{n}}\;\right\},\mathrm{and} \end{array}$$
(59)$$\begin{array}{c} N\left(n,M,P_{X}\right)=\min\;\left\{1,\alpha \left(n,M,P_{X}\right)\;+\;\frac{5\;\xi (P_{X})}{\sqrt{n}}\;+\;\frac{2\ln\left(2\right)}{\sigma {\left(P_{X}\right)}^{\frac{1}{2}}\sqrt{2n\pi }}\right\}, \end{array}$$
where
(60)$$\begin{array}{ccc} \alpha \left(n,M,P_{X}\right) & ≜ & Q\left(\frac{n\mu \left(P_{X}\right)-\ln\left(\frac{M-1}{2}\right)}{\sqrt{n\sigma (P_{X})}}\right). \end{array}$$

Using this notation, the following theorem introduces lower and upper bounds on the function *T* in (Equation 51).

**Theorem** **4.**
*Given a pair $\left(n,M\right)\in N^{2}$, for all input distributions $P_{X}\in ▵\left(X^{n}\right)$ subject to (Equation 53), the following holds with respect to the random transformation in (Equation 43) subject to (Equation 44),*
(61)$$\begin{array}{c} D\left(n,M,P_{X}\right)⩽T\left(n,M,P_{X}\right)⩽N\left(n,M,P_{X}\right), \end{array}$$
*where the functions T, D and N are defined in (51), (58) and (59), respectively.*


**Proof.** The proof of Theorem 4 is presented in [12]. Essentially, it relies on Theorem 1 for upper and lower bounding the terms $E_{P_{X}P_{Y|X}}\left[1_{\left\{\iota \left(X;Y\right)⩽\ln\left(\frac{M-1}{2}\right)\right\}}\right]$ in (Equation 51). The upper bound on $E_{P_{X}P_{Y}}\left[1_{\left\{\iota \left(X;Y\right)>\ln\left(\frac{M-1}{2}\right)\right\}}\right]$ in (Equation 51) follows from Lemma 47 in [10]. ☐

In [12], the function $\alpha (n,M,P_{X})$ in (Equation 60) is often referred to as the *normal approximation* of $T(n,M,P_{X})$, which is indeed a language abuse. In Section 2.1, a comment is given on the fact that the lower and upper bounds, i.e., the functions *D* in (Equation 58) and *N* in (59), are often too far from the normal approximation α in (Equation 60).

#### 3.2.2. Saddlepoint Approximation

This section describes an approximation of the function *T* in (Equation 51) by using the saddlepoint approximation of the CDF of the random variable ι(X;Y), as suggested in Section 2.2. Given a distribution $P_{X}\in ▵\left(X\right)$, the moment generating function of ι(X;Y) is
(62)$$\begin{array}{c} \varphi \left(P_{X},\theta \right)≜E_{P_{X}P_{Y|X}}\left[\exp\left(\theta \;\iota (X;Y)\right)\right], \end{array}$$
with θ∈R. For all $P_{X}\in ▵\left(X\right)$ and for all θ∈R, consider the following functions: (63)$$\begin{array}{c} \mu \left(P_{X},\theta \right)≜E_{P_{X}P_{Y|X}}\left[\frac{\iota \left(X;Y\right)\exp\left(\theta \;\iota (X;Y)\right)}{\varphi (P_{X},\theta )}\right], \end{array}$$(64)$$\begin{array}{c} V\left(P_{X},\theta \right)≜E_{P_{X}P_{Y|X}}\;\left[\;{\left(\iota \left(X;Y\right)\;-\;\mu \left(P_{X},\theta \right)\right)}^{2}\;\frac{\exp\left(\theta \;\iota (X;Y)\right)}{\varphi (P_{X},\theta )}\;\right],\mathrm{and} \end{array}$$(65)$$\begin{array}{c} \xi \left(P_{X},\theta \right)≜c_{1}\left(\frac{E_{P_{X}P_{Y|X}}\;\left[\;{\left|\iota \left(X;Y\right)\;-\;\mu \left(P_{X},\theta \right)\right|}^{3}\frac{\exp\left(\theta \iota (X;Y)\right)}{\varphi (P_{X},\theta )}\right]}{{\left(V(P_{X},\theta )\right)}^{3/2}}+c_{2}\right), \end{array}$$
where $c_{1}$ and $c_{2}$ are defined in (23). Using this notation, consider the functions $\beta_{1}:N^{2}\times R\times ▵\left(X\right)\to R_{+}$ and $\beta_{2}:N^{2}\times R\times ▵\left(X\right)\to R_{+}$:(66)$$\begin{array}{c} \beta_{1}\left(n,M,\theta ,P_{X}\right)\;\;=\;\;1_{\left\{\;\theta >0\;\right\}}\;+\;{\left(\;-\;1\;\right)}^{1_{\left\{\;\theta >0\;\right\}}}\exp\;\left(\;n\ln\left(\varphi (\;P_{X},\theta \;)\right)\;-\;\theta \ln\left(\;\frac{M\;-\;1}{2}\;\right)\;+\;\frac{1}{2}\theta^{2}nV\left(\;P_{X},\theta \;\right)\;\right)\;Q\;\left(\;\sqrt{nV(\;P_{X},\theta \;)}\left|\theta \right|\;\right),\;\; \end{array}$$
and
(67)$$\begin{array}{c} \beta_{2}\left(n,M,\theta ,P_{X}\right)\;\\ =1_{\left\{\;\theta ⩽\;-1\;\right\}}\;+\;{\left(\;-\;1\;\right)}^{1_{\left\{\theta ⩽\;-\;1\right\}}}\exp\;\left(\;n\ln\left(\;\varphi (\;P_{X},\;\theta \;)\;\right)\;-\;\left(\;\theta \;+\;1\;\right)\;\ln\left(\;\frac{M\;-\;1}{2}\;\right)\;+\;\frac{1}{2}{\left(\;\theta \;+\;1\;\right)}^{2}nV\;\left(\;P_{X},\;\theta \;\right)\;\right)Q\;\left(\;\sqrt{nV(\;P_{X},\theta \;)}\left|\;\theta \;+\;1\;\right|\;\right)\;. \end{array}$$

Note that $\beta_{1}$ is the saddlepoint approximation of the CDF of the random variable ι(X;Y) in (Equation 54) when X and Y follow the distribution $P_{X}P_{Y|X}$. Note also that $\beta_{2}$ is the saddlepoint approximation of the complementary CDF of the random variable ι(X;Y) in (Equation 54) when X and Y follow the distribution $P_{X}P_{Y}$.

Consider also the following functions: (68)$$\begin{array}{c} G_{1}\left(n,M,\theta ,P_{X}\right)=\beta_{1}\left(n,M,\theta ,P_{X}\right)-\frac{2\xi (P_{X},\theta )}{\sqrt{n}}\exp\;\left(\;n\ln\left(\varphi (P_{X},\theta )\right)-\theta \ln\left(\;\frac{M-1}{2}\;\right)\;\right), \end{array}$$(69)$$\begin{array}{c} G_{2}\left(n,M,\theta ,P_{X}\right)=\beta_{2}\left(n,M,\theta ,P_{X}\right)-\;\frac{2\xi (P_{X},\theta )}{\sqrt{n}}\exp\;\left(\;n\ln\left(\varphi (P_{X},\theta )\right)\;-\;\left(\;\theta \;+\;1\;\right)\ln\left(\;\frac{M\;-\;1}{2}\;\right)\;\right),\; \end{array}$$(70)$$\begin{array}{c} G\left(n,M,\theta ,P_{X}\right)=\max\left\{0,G_{1}\left(n,M,\theta ,P_{X}\right)\right\}+\frac{M-1}{2}\max\left\{0,G_{2}\left(n,M,\theta ,P_{X}\right)\right\},\;\mathrm{and} \end{array}$$(71)$$\begin{array}{c} S\left(n,M,\theta ,P_{X}\right)\;=\;\min\left\{1,\beta \left(n,M,\theta ,P_{X}\right)+\frac{4\xi (P_{X},\theta )}{\sqrt{n}}\exp\left(n\ln\left(\varphi (P_{X},\theta )\right)-\theta \ln\left(\frac{M-1}{2}\right)\right)\right\},\;\; \end{array}$$
where,
(72)$$\begin{array}{c} \beta \left(n,M,\theta ,P_{X}\right)=\beta_{1}\left(n,M,\theta ,P_{X}\right)+\frac{M-1}{2}\beta_{2}\left(n,M,\theta ,P_{X}\right),\; \end{array}$$
with $\beta_{1}$ in (Equation 66) and $\beta_{2}$ in (Equation 67). Often, the function β in (Equation 72) is referred to as the *saddlepoint approximation* of the function *T* in (Equation 51), which is indeed a language abuse.

The following theorem introduces new lower and upper bounds on the function *T* in (Equation 51).

**Theorem** **5.**
*Given a pair $\left(n,M\right)\in N^{2}$, for all input distributions $P_{X}\in ▵\left(X^{n}\right)$ subject to (Equation 53), the following holds with respect to the random transformation in (Equation 43) subject to (Equation 44),*
(73)$$\begin{array}{c} G\left(n,M,\theta ,P_{X}\right)⩽T\left(n,M,P_{X}\right)⩽S\left(n,M,\theta ,P_{X}\right) \end{array}$$
*where θ is the unique solution in t to*
(74)$$\begin{array}{c} n\mu \left(P_{X},t\right)=\ln\left(\frac{M-1}{2}\right), \end{array}$$

*and the functions T, G, and S are defined in (51), (70), and (71).*


**Proof.** The proof of Theorem 5 is provided in Appendix F. In a nutshell, the proof relies on Theorem 3 for independently bounding the terms $E_{P_{X}P_{Y|X}}\left[1_{\left\{\iota \left(X;Y\right)⩽\ln\left(\frac{M-1}{2}\right)\right\}}\right]$ and $E_{P_{X}P_{Y}}\left[1_{\left\{\iota \left(X;Y\right)>\ln\left(\frac{M-1}{2}\right)\right\}}\right]$ in (Equation 51). ☐

### 3.3. Meta Converse Bound

This section describes a lower bound on $\lambda^{*}\left(n,M\right)$, for a fixed pair $\left(n,M\right)\in N^{2}$. Given two probability distributions $P_{XY}\in ▵\left(X^{n}\times Y^{n}\right)$ and $Q_{Y}\in ▵\left(Y^{n}\right)$, let the random variable $\tilde{\iota }\left(X;Y|Q_{Y}\right)$ satisfy
(75)$$\tilde{\iota }\left(X;Y|Q_{Y}\right)≜\ln\left(\frac{dP_{XY}}{dP_{X}Q_{Y}}\left(X,Y\right)\right).$$

For all (n,*M*,γ)∈$N^{2}\times R$ and for all probability distributions $P_{X}\in ▵\left(X^{n}\right)$ and $Q_{Y}\in ▵\left(Y^{n}\right)$, let the function $C:N^{2}\times ▵\left(X^{n}\right)\times ▵\left(Y^{n}\right)\times R_{+}\to R_{+}$ be
(76)$$\begin{array}{c} C\left(n,M,P_{X},Q_{Y},\gamma \right)≜E_{P_{X}P_{Y|X}}\left[1_{\left\{\tilde{\iota }\left(X;Y|Q_{Y}\right)⩽\ln\left(\gamma \right)\right\}}\right]+\gamma \left(E_{P_{X}Q_{Y}}\left[1_{\left\{\tilde{\iota }\left(X;Y|Q_{Y}\right)>\ln\left(\gamma \right)\right\}}\right]-\frac{1}{M}\right).\; \end{array}$$

Using this notation, the following lemma describes the MC bound.

**Lemma** **3**(MC Bound [10,13]). *Given a pair (n,M)∈$N^{2}$, the following holds for all $Q_{Y}\in \Delta \left(Y^{n}\right)$, with respect to the random transformation in (Equation 43):*
(77)$$\begin{array}{c} \lambda^{*}\left(n,M\right)⩾\underset{P_{X}\in \Delta \left(X^{n}\right)}{\inf}\max_{\gamma ⩾0}C\left(n,M,P_{X},Q_{Y},\gamma \right), \end{array}$$
*where the function C is defined in (Equation 76).*


Note that the output probability distribution $Q_{Y}$ in Lemma 3 can be chosen among all possible probability distributions $Q_{Y}\in ▵\left(Y^{n}\right)$ to maximize the right-hand side of (Equation 76), which improves the bound. Note also that, with some loss of optimality, the optimization domain can be restricted to the set of probability distributions for which for all y∈$Y^{n}$,
(78)$$\begin{array}{c} Q_{Y}\left(y\right)=\prod_{t=1}^{n}Q_{Y}\left(y_{t}\right), \end{array}$$
with $Q_{Y}\in ▵\left(Y\right)$. Hence, subject to (Equation 44), for all $x\in X^{n}$, the random variable $\tilde{\iota }\left(x;Y|Q_{Y}\right)$ in (Equation 76) can be written as the sum of the independent random variables, i.e.,
(79)$$\begin{array}{c} \tilde{\iota }\left(x;Y|Q_{Y}\right)=\sum_{t=1}^{n}\tilde{\iota }\left(x_{t};Y_{t}|Q_{Y}\right). \end{array}$$

With some loss of generality, the focus is on a channel transformation of the form in (Equation 43) for which the following condition holds: The infimum in (Equation 77) is achieved by a product distribution, i.e., $P_{X}$ is of the form in (Equation 53), when the probability distribution $Q_{Y}$ satisfies (Equation 78). Note that this condition is met by memoryless channels such as the BSC, the AWGN and SαS channels with binary antipodal inputs, i.e., input alphabets are of the form X={a,−a}, with a∈R. This follows from the fact that the random variable $\tilde{\iota }\left(x;Y|Q_{Y}\right)$ is invariant of the choice of $x\in X^{n}$ when the probability distribution $Q_{Y}$ satisfies (Equation 78) and for all y∈Y,
(80)$$\begin{array}{c} Q_{Y}\left(y\right)=\frac{P_{Y|X}\left(y|-a\right)+P_{Y|X}\left(y|a\right)}{2}. \end{array}$$

Under these conditions, the random variable $\tilde{\iota }\left(X;Y|Q_{Y}\right)$ in (Equation 76) can be written as the sum of i.i.d. random variables, i.e.,
(81)$$\begin{array}{c} \tilde{\iota }\left(X;Y|Q_{Y}\right)=\sum_{t=1}^{n}\tilde{\iota }\left(X_{t};Y_{t}|Q_{Y}\right). \end{array}$$

This observation motivates the application of the results of Section 2 to provide upper and lower bounds on the function *C* in (Equation 76), for some given values $\left(n,M\right)\in N^{2}$ and given distributions $P_{X}\in ▵\left(X^{n}\right)$ and $Q_{Y}\in ▵\left(Y^{n}\right)$. These bounds become significantly relevant when the exact value of $C(n,M,P_{X},Q_{Y},\gamma )$ cannot be calculated with respect to the random transformation in (Equation 43). In such a case, providing upper and lower bounds on $C(n,M,P_{X},Q_{Y},\gamma )$ helps in approximating its exact value subject to an error sufficiently small such that the approximation is relevant.

#### 3.3.1. Normal Approximation

This section describes the normal approximation of the function *C* in (Equation 76), that is to say, the random variable $\tilde{\iota }\left(X;Y|Q_{Y}\right)$ is assumed to satisfy (Equation 81) and to follow a Gaussian distribution. More specifically, for all $\left(P_{X},Q_{Y}\right)\in ▵\left(X\right)\times ▵\left(Y\right)$, let
(82)$$\begin{array}{c} \tilde{\mu }\left(P_{X},Q_{Y}\right)≜E_{P_{X}P_{Y|X}}\left[\tilde{\iota }\left(X;Y|Q_{Y}\right)\right], \end{array}$$
(83)$$\begin{array}{c} \tilde{\sigma }\left(P_{X},Q_{Y}\right)≜E_{P_{X}P_{Y|X}}\left[{\left(\tilde{\iota }\left(X;Y|Q_{Y}\right)-\tilde{\mu }\left(P_{X},Q_{Y}\right)\right)}^{2}\right],\mathrm{and} \end{array}$$
(84)$$\begin{array}{c} \tilde{\xi }\left(P_{X},Q_{Y}\right)≜c_{1}\left(\frac{E_{P_{X}P_{Y|X}}\left[|\tilde{\iota }\left(X;Y|Q_{Y}\right)-\tilde{\mu }\left(P_{X},Q_{Y}\right){|}^{3}\right]}{{\left(\tilde{\sigma }\left(P_{X},Q_{Y}\right)\right)}^{3/2}}+c_{2}\right) \end{array}$$
with $c_{1}$ and $c_{2}$ defined in (23), be functions of the input and output distributions $P_{X}$ and $Q_{Y}$, respectively. In particular, $\tilde{\mu }\left(P_{X},Q_{Y}\right)$ and $\tilde{\sigma }\left(P_{X},Q_{Y}\right)$ are respectively the first moment and the second central moment of the random variables $\tilde{\iota }\left(X_{1};Y_{1}|Q_{Y}\right),\tilde{\iota }\left(X_{2};Y_{2}|Q_{Y}\right),\ldots \tilde{\iota }\left(X_{n};Y_{n}|Q_{Y}\right)$. Using this notation, consider the functions $\tilde{D}:N^{2}\times ▵\left(X\right)\times ▵\left(Y\right)\times R_{+}\to R_{+}$ and $\tilde{N}:N^{2}\times ▵\left(X\right)\times ▵\left(Y\right)\times R_{+}\to R_{+}$ such that, for all $\left(n,M,\gamma \right)\in N^{2}\times R_{+}$ and for all $P_{X}\in ▵\left(X\right)$ and for all $Q_{Y}\in ▵\left(Y\right)$,
(85)$$\begin{array}{c} \tilde{D}\left(n,M,P_{X},Q_{Y},\gamma \right)=\max\;\left\{\;0,\tilde{\alpha }\;\left(\;n,M,P_{X},Q_{Y},\gamma \right)\;-\;\frac{\tilde{\xi }\left(P_{X},Q_{Y}\right)}{\sqrt{n}}\;\right\},\mathrm{and} \end{array}$$
(86)$$\begin{array}{c} \tilde{N}\left(n,M,P_{X},Q_{Y},\gamma \right)=\min\;\left\{\;1,\;\tilde{\alpha }\left(n,M,P_{X},Q_{Y},\gamma \right)\;+\;\frac{5\;\tilde{\xi }\left(P_{X},Q_{Y}\right)}{\sqrt{n}}\;+\;\frac{2\ln\left(2\right)}{\tilde{\sigma }{\left(P_{X},Q_{Y}\right)}^{\frac{1}{2}}\sqrt{2n\pi }}\;\right\},\; \end{array}$$
where
(87)$$\begin{array}{ccc} \tilde{\alpha }\left(n,M,P_{X},Q_{Y},\gamma \right) & ≜ & Q\left(\frac{n\tilde{\mu }\left(P_{X},Q_{Y}\right)-\ln\left(\gamma \right)}{\sqrt{n\tilde{\sigma }\left(P_{X},Q_{Y}\right)}}\right)-\frac{\gamma }{M}. \end{array}$$

Using this notation, the following theorem introduces lower and upper bounds on the function *C* in (Equation 76).

**Theorem** **6.**
*Given a pair $\left(n,M\right)\in N^{2}$, for all input distributions $P_{X}\in ▵\left(X^{n}\right)$ subject to (Equation 53), for all output distributions $Q_{Y}\in ▵\left(Y^{n}\right)$ subject to (Equation 78), and for all γ⩾0, the following holds with respect to the random transformation in (Equation 43) subject to (Equation 44),*
(88)$$\begin{array}{c} \tilde{D}\left(n,M,P_{X},Q_{Y},\gamma \right)⩽C\left(n,M,P_{X},Q_{Y},\gamma \right)⩽\tilde{N}\left(n,M,P_{X},Q_{Y},\gamma \right), \end{array}$$
*where the functions C, $\tilde{D}$, and $\tilde{N}$ are defined in (Equation 76), (Equation 85), and (86), respectively.*


**Proof.** The proof of Theorem 6 is partially presented in [10]. Essentially, it relies on Theorem 1 for upper and lower bounding the term $E_{P_{X}P_{Y|X}}\left[1_{\left\{\tilde{\iota }\left(X;Y|Q_{Y}\right)⩽\ln\left(\gamma \right)\right\}}\right]$ in (Equation 76); and using Lemma 47 in [10] for upper bounding the term $E_{P_{X}Q_{Y}}\left[1_{\left\{\tilde{\iota }\left(X;Y|Q_{Y}\right)>\ln\left(\gamma \right)\right\}}\right]$ in (Equation 76). ☐

The function $\tilde{\alpha }\;\left(\;n,M,P_{X},Q_{Y},\gamma \right)$ in (Equation 87) is often referred to as the *normal approximation* of $C(n,M,P_{X})$, which is indeed a language abuse. In Section 2.1, a comment is given on the fact that the lower and upper bounds on the normal approximation, i.e., the functions $\tilde{D}$ in (Equation 85) and $\tilde{N}$ in (86), are often too far from the normal approximation $\tilde{\alpha }$ in (Equation 87).

#### 3.3.2. Saddlepoint Approximation

This section describes an approximation of the function *C* in (Equation 76) by using the saddlepoint approximation of the CDF of the random variable $\tilde{\iota }\left(X;Y|Q_{Y}\right)$, as suggested in Section 2.2. Given two distributions $P_{X}\in ▵\left(X\right)$ and $Q_{Y}\in ▵\left(Y\right)$, let the random variable $\tilde{\iota }\left(X;Y|Q_{Y}\right)$ satisfy
(89)$$\begin{array}{c} \tilde{\iota }\left(X;Y|Q_{Y}\right)≜\ln\left(\frac{dP_{X}P_{Y|X}}{dP_{X}Q_{Y}}\left(X,Y\right)\right), \end{array}$$
where $P_{Y|X}$ is in (Equation 44). The moment generating function of $\tilde{\iota }\left(X;Y|Q_{Y}\right)$ is
(90)$$\begin{array}{c} \tilde{\varphi }\left(P_{X},Q_{Y},\theta \right)≜E_{P_{X}P_{Y|X}}\left[\exp\left(\theta \;\tilde{\iota }\left(X;Y|Q_{Y}\right)\right)\right], \end{array}$$
with θ∈R. For all $P_{X}\in ▵\left(X\right)$ and $Q_{Y}\in ▵\left(Y\right)$, and for all θ∈R, consider the following functions: (91)$$\begin{array}{c} \tilde{\mu }\left(P_{X},Q_{Y},\theta \right)≜E_{P_{X}P_{Y|X}}\left[\frac{\tilde{\iota }\left(X;Y|Q_{Y}\right)\exp\left(\theta \;\tilde{\iota }\left(X;Y|Q_{Y}\right)\right)}{\tilde{\varphi }\left(P_{X},Q_{Y},\theta \right)}\right], \end{array}$$(92)$$\begin{array}{c} \tilde{V}\left(P_{X},Q_{Y},\theta \right)≜E_{P_{X}P_{Y|X}}\;\left[\;{\left(\tilde{\iota }\left(X;Y|Q_{Y}\right)\;-\;\tilde{\mu }\left(P_{X},Q_{Y},\theta \right)\right)}^{2}\;\frac{\exp\left(\theta \;\tilde{\iota }\left(X;Y|Q_{Y}\right)\right)}{\tilde{\varphi }\left(P_{X},Q_{Y},\theta \right)}\;\right]\;,\mathrm{and} \end{array}$$(93)$$\begin{array}{c} \tilde{\xi }\left(P_{X},Q_{Y},\theta \right)≜c_{1}\left(\frac{E_{P_{X}P_{Y|X}}\;\left[\;{\left|\tilde{\iota }\left(X;Y|Q_{Y}\right)\;-\;\tilde{\mu }\left(P_{X},Q_{Y},\theta \right)\right|}^{3}\frac{\exp\left(\theta \tilde{\iota }\left(X;Y|Q_{Y}\right)\right)}{\tilde{\varphi }\left(P_{X},Q_{Y},\theta \right)}\right]}{{\left(\tilde{V}\left(P_{X},Q_{Y},\theta \right)\right)}^{3/2}}+c_{2}\right)\;, \end{array}$$
where $c_{1}$ and $c_{2}$ are defined in (23). Using this notation, consider the functions ${\tilde{\beta }}_{1}:N\times R_{+}\times R\times ▵\left(X\right)\times ▵\left(Y\right)\to R_{+}$ and ${\tilde{\beta }}_{2}:N\times R_{+}\times R\times ▵\left(X\right)\times ▵\left(Y\right)\to R_{+}$: $$\begin{array}{c} {\tilde{\beta }}_{1}\left(n,\gamma ,\theta ,P_{X},Q_{Y}\right) \end{array}$$(94)$$\begin{array}{c} =\;1_{\left\{\;\theta >0\;\right\}}\;+\;{\left(\;-1\;\right)}^{1_{\left\{\;\theta >0\;\right\}}}\;\exp\;\left(\;n\ln\left(\;\tilde{\varphi }\left(P_{X},Q_{Y},\theta \right)\;\right)\;-\;\theta \ln\left(\gamma \right)\;+\;\frac{1}{2}\theta^{2}n\tilde{V}\left(\;P_{X},\;Q_{Y},\;\theta \right)\;\right)\;Q\;\left(\;\sqrt{\;n\tilde{V}\left(P_{X},Q_{Y},\theta \right)\;}\left|\theta \right|\;\right),\mathrm{and}\;\\ {\tilde{\beta }}_{2}\left(n,\gamma ,\theta ,P_{X},Q_{Y}\right)\;\\ =\;1_{\left\{\;\theta ⩽\;-1\;\right\}}\;+\;{\left(\;-\;1\;\right)}^{1_{\left\{\theta ⩽\;-\;1\right\}}}\exp\left(\;n\ln\left(\tilde{\varphi }\left(P_{X},Q_{Y},\theta \right)\right)\;-\;\left(\;\theta \;+\;1\;\right)\;\ln\left(\gamma \right)\;+\;\frac{1}{2}{\left(\;\theta \;+\;1\;\right)}^{2}n\tilde{V}\left(\;P_{X},Q_{Y},\theta \;\right)\;\right)\; \end{array}$$(95)$$\begin{array}{c} Q\;\left(\;\sqrt{n\tilde{V}\left(\;P_{X},Q_{Y},\theta \;\right)}\left|\;\theta \;+\;1\;\right|\;\right)\;. \end{array}$$

Note that ${\tilde{\beta }}_{1}$ and ${\tilde{\beta }}_{2}$ are the saddlepoint approximation of the CDF and the complementary CDF of the random variable $\tilde{\iota }\left(X;Y|Q_{Y}\right)$ in (Equation 81) when $\left(X,Y\right)$ follows the distribution $P_{X}P_{Y|X}$ and $P_{X}Q_{Y}$, respectively. Consider also the following functions: (96)$$\begin{array}{c} {\tilde{G}}_{1}\left(n,\gamma ,\theta ,P_{X},Q_{Y}\right)={\tilde{\beta }}_{1}\left(n,\gamma ,\theta ,P_{X},Q_{Y}\right)-\frac{2\tilde{\xi }\left(P_{X},Q_{Y},\theta \right)}{\sqrt{n}}\exp\left(n\ln\left(\tilde{\varphi }\left(P_{X},Q_{Y},\theta \right)\right)-\theta \ln\left(\gamma \right)\right), \end{array}$$(97)$$\begin{array}{c} {\tilde{G}}_{2}\left(n,\gamma ,\theta ,P_{X},Q_{Y}\right)={\tilde{\beta }}_{2}\left(n,\gamma ,\theta ,P_{X},Q_{Y}\right)-\frac{2\tilde{\xi }\left(P_{X},Q_{Y},\theta \right)}{\sqrt{n}}\exp\left(n\ln\left(\tilde{\varphi }\left(P_{X},Q_{Y},\theta \right)\right)-\left(\theta +1\right)\ln\left(\gamma \right)\right),\; \end{array}$$(98)$$\begin{array}{c} \tilde{G}\left(n,\gamma ,\theta ,P_{X},Q_{Y},M\right)=\max\left\{0,{\tilde{G}}_{1}\left(n,\gamma ,\theta ,P_{X},Q_{Y}\right)\right\}+\gamma \max\left\{0,{\tilde{G}}_{2}\left(n,\gamma ,\theta ,P_{X},Q_{Y}\right)\right\}-\frac{\gamma }{M}, \end{array}$$(99)$$\begin{array}{c} \tilde{S}\left(n,\gamma ,\theta ,P_{X},Q_{Y},M\right)\;=\;\min\;\left\{\;1,\tilde{\beta }\left(n,\gamma ,\theta ,P_{X},Q_{Y},M\right)\;+\;\frac{4\tilde{\xi }\left(P_{X},Q_{Y},\theta \right)}{\sqrt{n}}\exp\;\left(\;n\ln\left(\;\tilde{\varphi }\left(P_{X},Q_{Y},\theta \right)\;\right)\;-\;\theta \ln\left(\;\gamma \;\right)\;\right)\;\right\}\;, \end{array}$$
and
(100)$$\tilde{\beta }\left(n,\gamma ,\theta ,P_{X},Q_{Y},M\right)={\tilde{\beta }}_{1}\left(n,\gamma ,\theta ,P_{X},Q_{Y}\right)+\gamma {\tilde{\beta }}_{2}\left(n,\;\gamma ,\theta ,P_{X},Q_{Y}\right)-\frac{\gamma }{M}.$$

The function $\tilde{\beta }\left(n,\gamma ,\theta ,P_{X},Q_{Y},M\right)$ in (Equation 100) is referred to as the *saddlepoint approximation* of the function *C* in (Equation 76), which is indeed a language abuse.

The following theorem introduces new lower and upper bounds on the function *C* in (Equation 76).

**Theorem** **7.**
*Given a pair $\left(n,M\right)\in N^{2}$, for all input distributions $P_{X}\in ▵\left(X^{n}\right)$ subject to (Equation 53), for all output distributions $Q_{Y}\in ▵\left(Y^{n}\right)$ subject to (Equation 81) such that for all x∈X, $P_{Y|X=x}$ is absolutely continuous with respect to $Q_{Y}$, for all γ⩾0, the following holds with respect to the random transformation in (Equation 43) subject to (Equation 44),*
(101)$$\begin{array}{c} \tilde{G}\left(n,\gamma ,\theta ,P_{X},Q_{Y},M\right)⩽C\left(n,M,P_{X},Q_{Y},\gamma \right)⩽\tilde{S}\left(n,\gamma ,\theta ,P_{X},Q_{Y},M\right) \end{array}$$
*where θ is the unique solution in t to*
(102)$$\begin{array}{c} n\mu \left(P_{X},t\right)=\ln\left(\gamma \right), \end{array}$$

*and the functions C, $\tilde{G}$, and $\tilde{S}$ are defined in (76), (98) and (99).*


**Proof.** The proof of Theorem 7 is provided in Appendix G. ☐

Note that, in (Equation 101), the parameter γ can be optimized as in (Equation 77).

### 3.4. Numerical Experimentation

The normal and the saddlepoint approximations of the DT and MC bounds as well as their corresponding upper and lower bounds presented from Section 3.2.1 to Section 3.3.2 are studied in the cases of the BSC, the AWGN channel, and the SαS channel. The latter is defined by the random transformation in (Equation 43) subject to (Equation 44) and for all (x,y)∈X×Y:(103)$$\begin{array}{c} P_{Y|X}\left(y|x\right)=P_{Z}\left(y-x\right), \end{array}$$
where $P_{Z}$ is a probability distribution satisfying for all t∈R,
(104)$$\begin{array}{c} E_{P_{Z}}\left[\exp\left(itZ\right)\right]=\exp\left(-{\left|\sigma t\right|}^{\alpha }\right), \end{array}$$
with $i=\sqrt{-1}$. The reals α∈(0,2] and $\sigma \in R_{+}$ in (Equation 104) are parameters of the SαS channel.

In the following figures, Figure 3, Figure 4 and Figure 5, the channel inputs are discrete X={−1,1}, $P_{X}$ is the uniform distribution, and θ is chosen to be the unique solution to *t* in (Equation 74) or (Equation 102) depending on whether the DT or MC bound is considered. For the results relative to the MC bound, $Q_{Y}$ is chosen to be equal to the distribution $P_{Y}$, i.e., the marginal of $P_{X}P_{Y|X}$. The parameter γ is chosen to maximize the function $C(n,2^{nR},P_{X},Q_{Y},\gamma )$ in (Equation 76). The plots in Figure 3a, Figure 4a and Figure 5a illustrate the function $T(n,2^{nR},P_{X})$ in (Equation 51) as well as the bounds in Theorems 4 and 5. Figure 3b, Figure 4b and Figure 5b illustrate the function *C* in (Equation 76) and the bounds in Theorems 6 and 7. The normal approximations, i.e, $\alpha \left(n,2^{nR},P_{X}\right)$ in (Equation 60) and $\tilde{\alpha }\left(n,2^{nR},P_{X},Q_{Y},\gamma \right)$ in (Equation 87), of the DT and MC bounds, respectively, are plotted in black diamonds. The upper bounds, i.e., $N\left(n,2^{nR},P_{X}\right)$ in (59) and $\tilde{N}\left(n,2^{nR},P_{X},Q_{Y},\gamma \right)$ in (86), are plotted in blue squares. The lower bounds of the DT and MC bounds, i.e., $D\left(n,M,P_{X}\right)$ in (Equation 58) and $\tilde{D}\left(n,2^{nR},P_{X},Q_{Y},\gamma \right)$ in (Equation 85), are non-positive in these cases, and thus do not appear in the figures. The saddlepoint approximations of the DT and MC bounds, i.e., $\beta \left(n,2^{nR},\theta ,P_{X}\right)$ in (Equation 72) and $\tilde{\beta }\left(n,\gamma ,\theta ,P_{X},Q_{Y},2^{nR}\right)$ in (Equation 100), respectively, are plotted in black stars. The upper bounds, i.e., $S\left(n,2^{nR},\theta ,P_{X}\right)$ in (71) and $\tilde{S}\left(n,\gamma ,\theta ,P_{X},Q_{Y},2^{nR}\right)$ in (99), are plotted in blue upward-pointing triangles. The lower bounds, i.e., $G\left(n,2^{nR},\theta ,P_{X}\right)$ in (70) and $\tilde{G}\left(n,\gamma ,\theta ,P_{X},Q_{Y},2^{nR}\right)$ in (98), are plotted in red downward-pointing triangles.

Figure 3 illustrates the case of a BSC with cross-over probability δ=0.11. The information rates are chosen to be R=0.32 and R=0.42 bits per channel use in Figure 3a,b, respectively. The functions *T* and *C* can be calculated exactly and thus they are plotted in magenta asterisks in Figure 3a,b, respectively. In these figures, it can be observed that the saddlepoint approximations of the DT and MC bounds, i.e., β and $\tilde{\beta }$, respectively, overlap with the functions *T* and *C*. These observations are in line with those reported in [13]. Therein, the saddlepoint approximations of the RCU bound and the MC bound are both shown to be precise approximations. Alternatively, the normal approximations of the DT and MC bounds, i.e., α and $\tilde{\alpha }$, do not overlap with *T* and *C* respectively.

In Figure 3, it can be observed that the new bounds on the DT and MC provided in Theorems 5 and 7, respectively, are tighter than those in Theorems 4 and 6. Indeed, the upper-bounds *N* and $\tilde{N}$ on the DT and MC bounds derived from the normal approximations α and $\tilde{\alpha }$, are several order of magnitude above *T* and *C*, respectively. This observation remains valid for AWGN channels in Figure 4 and SαS channels in Figure 5, respectively. Note that, in Figure 3a, for n>1000, the normal approximation α is below the lower bound *G* showing that approximating *T* by α is too optimistic. These results show that the use of the Berry–Esseen Theorem to approximate the DT and MC bounds may lead to erroneous conclusions due to the uncontrolled error made on the approximation.

Figure 4 and Figure 5 illustrate the cases of a real-valued AWGN channel and a SαS channel, respectively. The signal-to-noise ratio (SNR) is SNR=1 for the AWGN channel. The information rate is R=0.425 bits per channel use for the AWGN channel and R=0.38 bits per channel use for the SαS channel with (α,σ)=(1.4,0.6). In both cases, the functions *T* in (Equation 51) and *C* in (Equation 76) can not be computed explicitly and hence does not appear in Figure 4 and Figure 5. In addition, the lower bounds $D\left(n,M,P_{X}\right)$ and $\tilde{D}\left(n,2^{nR},P_{X},Q_{Y},\gamma \right)$ obtained from Theorems 4 and 6 are non-positive in these cases, and thus, do not appear on these figures.

In Figure 4, note that the saddlepoint approximations, β and $\tilde{\beta }$, are well bounded by Theorems 5 and 7 for a large range of blocklengths. Alternatively, the lower bounds *D* and $\tilde{D}$ based on the normal approximation do not even exist in that case.

In Figure 5, note that the upper bounds *S* and $\tilde{S}$ on the DT and MC respectively are relatively tight compared to those in AWGN channel case. This characteristic is of a particular importance in a channel such as SαS channel, where the DT and MC bounds remain computable only by Monte Carlo simulations.

## 4. Discussion and Further Work

One of the main results of this work is Theorem 3, which gives an upper bound on the error induced by the saddlepoint approximation of the CDF of a sum of i.i.d. random variables. This result paves the way to study channel coding problems at any finite blocklength and any constraint on the DEP. In particular, Theorem 3 is used to bound the DT and MC bounds in point-to-point memoryless channels. This leads to tighter bounds than those obtained from Berry–Esseen Theorem (Theorem 1), cf., examples in Section 3.4, particularly for the small values of the DEP.

The bound on the approximation error presented in Theorem 2 uses a triangle inequality in the proof of Lemma A1, which is loose. This is essentially the reason why Theorem 2 is not reduced to the Berry–Esseen Theorem when the parameter θ is equal to zero. An interesting extension of this work is to tighten the inequality in Lemma A1 such that the Berry–Esseen Theorem can be obtained as a special case of Theorem 2, i.e., when θ=0. If such improvement on Theorem 2 is possible, Theorem 3 will be significantly improved and it would be more precise everywhere and in particular in the vicinity of the mean of the sum in (Equation 1). 

## Figures and Tables

**Figure 1 entropy-22-00690-f001:**
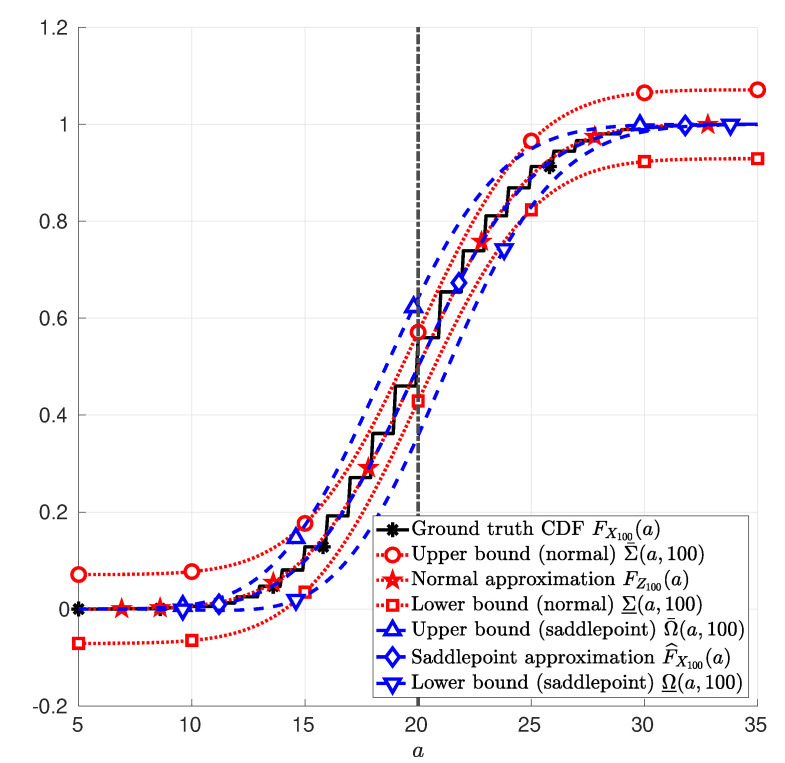
Sum of 100 Bernoulli random variables with parameter p=0.2. The function $F_{X_{100}}\left(a\right)$ (asterisk markers *) in Example 1; the function $F_{Z_{100}}\left(a\right)$ (star markers ⋆) in (Equation 25); the function ${\hat{F}}_{X_{100}}\left(a\right)$ (diamond markers ⋄) in (Equation 12); the function $\bar{\Sigma }\left(a,100\right)$ (circle marker ∘) in (Equation 26); the function $\underline{\Sigma }\left(a,100\right)$ (square marker □) in (27); the function $\bar{\Omega }\left(a,100\right)$ (upward-pointing triangle marker ▵) in (Equation 41); and the function $\underline{\Omega }\left(a,100\right)$ (downward-pointing triangle marker ▿) in (42) are plotted as functions of *a*, with a∈[5,35].

**Figure 2 entropy-22-00690-f002:**
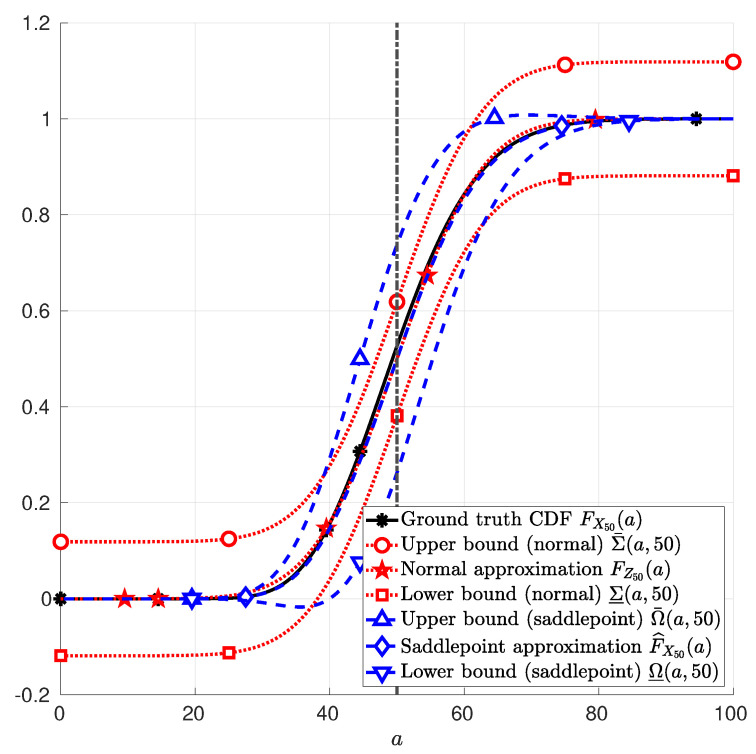
Sum of 50 Chi-squared random variables with parameter k=1. The function $F_{X_{50}}\left(a\right)$ (asterisk markers *) in Example 2; the function $F_{Z_{50}}\left(a\right)$ (star markers ⋆) in (Equation 25); the function ${\hat{F}}_{X_{50}}\left(a\right)$ (diamond markers ⋄) in (Equation 12); the function $\bar{\Sigma }\left(a,50\right)$ (circle marker ∘) in (Equation 26); the function $\underline{\Sigma }\left(a,50\right)$ (square marker □) in (27); the function $\bar{\Omega }\left(a,50\right)$ (upward-pointing triangle marker ▵) in (Equation 41); and the function $\underline{\Omega }\left(a,50\right)$ (downward-pointing triangle marker ▿) in (42) are plotted as functions of *a*, with a∈[0,100].

**Figure 3 entropy-22-00690-f003:**
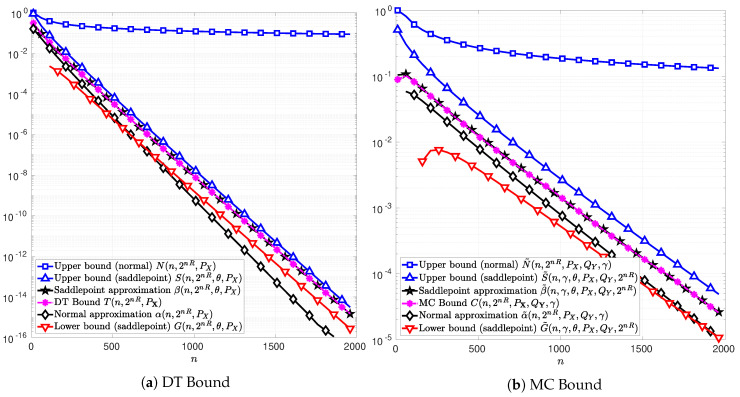
Normal and saddlepoint approximations to the functions *T* (Figure 3a) in (Equation 51) and *C* (Figure 3b) in (Equation 76) as functions of the blocklength *n* for the case of a BSC with cross-over probability δ=0.11. The information rate is R=0.32 and R=0.42 bits per channel use for Figure 3a,b, respectively. The channel input distribution $P_{X}$ is chosen to be the uniform distribution, the output distribution $Q_{Y}$ is chosen to be the channel output distribution $P_{Y}$, and the parameter γ is chosen to maximize *C* in (Equation 76). The parameter θ is chosen to be respectively the unique solution to *t* in (Equation 74) in Figure 3a and in (Equation 102) in Figure 3b.

**Figure 4 entropy-22-00690-f004:**
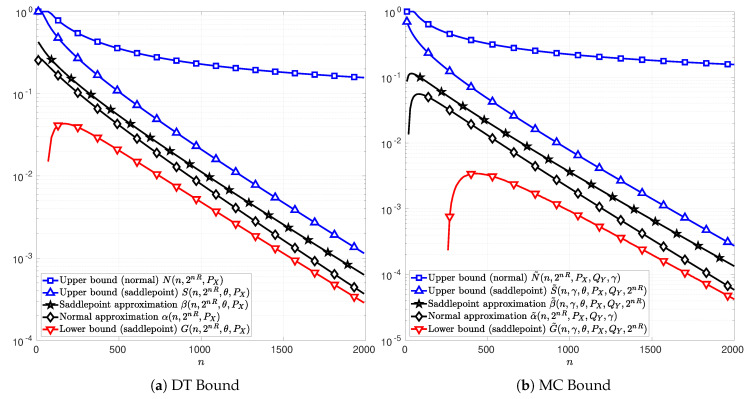
Normal and saddlepoint approximations to the functions *T* (Figure 4a) in (Equation 51) and *C* (Figure 4b) in (Equation 76) as functions of the blocklength *n* for the case of a real-valued AWGN channel with discrete channel inputs, X={−1,1}, signal to noise ratio SNR=1, and information rate R=0.425 bits per channel use. The channel input distribution $P_{X}$ is chosen to be the uniform distribution, the output distribution $Q_{Y}$ is chosen to be the channel output distribution $P_{Y}$, and the parameter γ is chosen to maximize *C* in (Equation 76). The parameter θ is respectively chosen to be the unique solution to *t* in (Equation 74) in Figure 4a and in (Equation 102) in Figure 4b.

**Figure 5 entropy-22-00690-f005:**
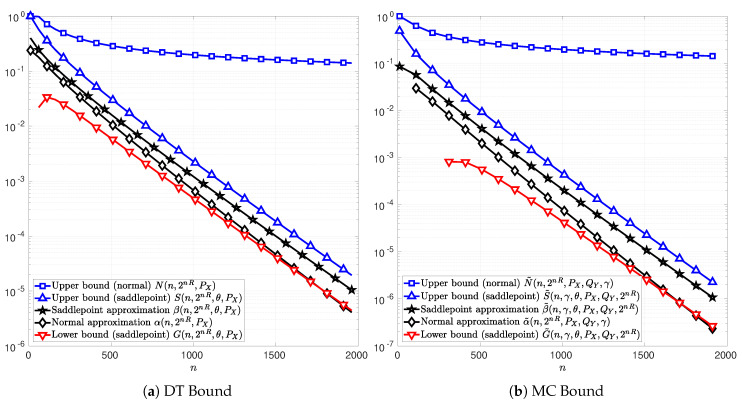
Normal and saddlepoint approximation to the functions *T* (Figure 5a) in (Equation 51) and *C* (Figure 5b) in (Equation 76) as functions of the blocklength *n* for the case of a real-valued symmetric α-stable noise channel with discrete channel inputs X={−1,1}, shape parameter α=1.4, dispersion parameter σ=0.6, and information rate R=0.38 bits per channel use. The channel input distribution $P_{X}$ is chosen to be the uniform distribution, the output distribution $Q_{Y}$ is chosen to be the channel output distribution $P_{Y}$, and the parameter γ is chosen to maximize *C* in (Equation 76). The parameter θ is respectively chosen to be the unique solution to *t* in (Equation 74) in Figure 5a and in (Equation 102) in Figure 5b.

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
