# Peer review of "An Upper Bound on the Error Induced by Saddlepoint Approximations—Applications to Information Theoryâ€"

_entropy, 2020, doi:10.3390/e22060690_

Round 1
Reviewer 1 Report
After reading the paper, I think, it can be given on the revision. The paper can be considered for the publication if the authors incorporate the following suggestions in the revised paper.
- Some concluding remarks and advantages of the proposed methods should be mentioned in the introduction.
- The introduction section should be started from the importance of the entropy, applications, and advantages.
- The gap and contribution of the paper are not clear.
- Why approximating the cumulative distribution function is considered? What is logic? An example should be added.
- A separate real example section should be added. The existing and current methods should be compared for the real example.
- A separate comparison section using the simulation study should be added.
- For the future study, the proposed method can be extended using the neutrosophic statistics. The following papers can be refereed in the revised paper in this regard
a) The W/S Test for Data having Neutrosophic Numbers-an Application to USA Villages Population
b) On Detecting Outliers in Complex Data Using Dixon’s, test Under Neutrosophic Statistics
c) Introducing Kolmogorov–Smirnov Tests under Uncertainty-an Application to Radioactive Data
d)New Diagnosis Test under the Neutrosophic Statistics-An Application to Diabetic Patients
e)Design of the Bartlett and Hartley Tests for Homogeneity of Variances under Indeterminacy Environment
f) Acceptance Sampling Plans for Two-Stage Process for Multiple Lines under Neutrosophic Statistics
8. Please update the reference list
Author Response
Dear Editor,
In the file attached to this message you will find:
- A document that answers all and each of the comments from the reviewers.
- The new version of the manuscript in which editions are marked in blue.
Best wishes,
Samir Perlaza
On behalve of the authors.

Reviewer 2 Report
This paper obtains an upper bound on the difference between the cumulative distribution function (CDF) of a sum of a finite number of i.i.d. random variables with finite absolute third moment, and the saddle-point approximation of this CDF. The requirement of finite absolute third moment needs to be mentioned explicitly in the abstract and introduction. The analysis relies on the Berry-Essen theorem, with the refined analysis of constants in [11]. The bound in Theorem 3 is then applied to study the minimum average decoding error probability for finite block length when the communication takes place over point-to-point memoryless channels, by obtaining upper and lower bounds on the dependence-test (DT) and meta-converse (MC) bounds in the paper [12] (Polyanskiy, Poor and Verdu). I find this work solid, and I have several suggestions:
1) Please incorporate the proof of Theorem 5 in the paper. It appears in Appendix F of the internal technical report [18], but it should also appear in the paper. I do not see a reason, especially in case of papers which are open access and without page limitations, not to include some of the proofs. The internal report may not be available at some point, while the paper should be. There is no need to refer to [18], once the proof of Theorem 5 is incorporated in the text (is there something else in [18] which does not appear in the submission ?).
2) You keep referring to the constant c = 0.476 in the Berry-Essen bound, while saying that this constant is according to the arXiv paper [11]. Looking at [11], there is no constant there which is equal to 0.476. I guess that you meant to refer to Corollary 1 in [11], where the suggested constant for the Berry-Essen bound is equal to 0.4748. But more importantly, Theorem 2 in [11] seems to suggest a better bound than Corollary 1 (although I should miss the reason why \beta_3 \geq 1 in [11]), which could be also used for further tightening your bound. Please clarify this point, and also the selection of your constant c throughout the paper.
3) Throughout the paper, \min( , ) should be written as \min \{ , \} (please change the type of parenthesis).
The following are minor suggestions (partially, also related to the English):
1) numerical analysis of these bounds: change 'analysis' to 'experimentation'.
2) Line 10: 'in which the new bounds' ---> 'where the new bounds'.
3) The constants a and b in Line 16 depend on n, so please change a and b to a_n and b_n respectively. Also, in Line 17, it should be "and Y is a random variable whose CDF is F_Y" (the first 'is' was missing).
4) Line 18: please replace to "the Gaussian, Cauchy or Levy distributions".
5) Line 26: complex numbers --> complex plane.
6) Two lines before (7): An expansion in Taylor series --> A Taylor series expansion.
7) The first line in Section 1.1 (paper contributions) it too complicated; to ease readability, please delete "of the CDF F_{X_n} of the sum in (1)." This is not needed, given what you wrote earlier, and it complicates the sentence for no good reason.
8) The three-lines sentence coming after Line 85 is redundant as it repeats in words what is written in (27). Please delete it.
9) Line 124, please change to "the DT and MC bounds".
10) 4th line of page 10:
lost --> loss, constrained --> restricted.
11) Line afterwards:
please add 'product' before 'probability distributions'.
12) Line 137: please write 'lower and upper bounds' (instead of 'a lower bound and an upper bound').
a lower bound on T is, according to (48), a lower bound on an upper bound on the minimal error probability with a fixed block length and rate. The only use I can see for this lower bound is to have an indication on the tightness of the upper bound on T. Please clarify this point in the paper.
13) Line 139: it consists in using --> it relies on
14) put more space or \cdot between the exponent and \exp, to improve readability.
15) There is no need for 'and' in (24), (29), (35), (52), (54), (60) etc.
16) Line 154: consists in using --> relies on
17) \gamma in (72) should be positive so \Reals should be \Reals^+ (two lines before (72)).
18) Sometimes you refer to [10], and sometimes to [17]. Is there anything relevant in [17] which is not included in [10] ?
Author Response
Dear Editor,
In the file attached to this message you will find:
A document that answers all and each of the comments from the reviewers.
The new version of the manuscript in which editions are marked in blue.
Best wishes,
Samir Perlaza
On behalve of the authors.

Round 2
Reviewer 1 Report
I read the reply to the following questions. I am not satisfied with the answers.
Answer to Comment 1.7: The authors went through the articles mentioned by the reviewer. In our
work, the statistics of the random variables Y1, Y2, . . . , Yn are perfectly determined and the objective is
to study the cumulative distribution function of their sum Xn = Y1 + Y2 + . . . + Yn for a fixed number
n. It may be interesting to study this cumulative distribution function by having an n that is not fixed
but belongs to an interval like in Neutrosophic Statistics. However, this is not the aim and the scope of
the paper. In consequence, the authors did not include these references.
Comment 1.8: Please update the reference list
Answer to Comment 1.8: We kindly ask to the reviewer to refer to our previous answer.
Additional comments: It is clear that the authors did not work on neutrosophic statistics and the paper was evaluated on the basis of classical statistics. As the authors agreed that n belongs to an interval is an interesting area. It was suggested to add the following line in the conclusion section.
The present study can be extended using neutrosophic statistics as future research. The reader may refer to the suggested references in the first round. Also, the authors did not update the reference list as suggested.
Author Response
Dear Editor and reviewers,
In the pdf file attached to this message you will find the answer to all comments from the reviewers.
Best wishes,
The authors
Reviewer 2 Report
My comments in the first round have been properly addressed in the current revised version. The introduction has also been improved.
I have, at this stage of revision, several minor comments:
1) The sentence in Line 8 reads better by writing: "include, respectively, new upper and lower bounds on the DT and MC bounds."
2) Line 63: please change the 'main results' to 'main objectives' (in regard to items (a) and (b) in lines (63)-(66)).
3) Line 67: please delete 'the case of'.
4) Second line on page 6: please modify 'consists in' to 'gives'; the same also applies to the line after 107.
5) Line 167 and the first line on page 31: replace 'consists in using' to 'relies on'
6) Line 257: please change 'strongly improved' to 'significantly improved'
7) Lines 297 and 310: please change 'The next step consists in simplifying' to 'The next step simplifies'.
8) Line after line 309: change to 'the next step derives';
9) First line of Appendix F: please change it to 'For a fixed product probability input distribution $P_X$ in (52) and for a memoryless transition probability in (43), the upper bound ….
10) it would be nicer to define once for all c_1 = 0.33554, and c_2 = 0.415, and then to replace the 18 times where each one of the constants 0.33554 and 0.415 appear in the paper by c_1 and c_2, respectively.
11) Some acronyms are defined in the body of the paper (after the abstract) more than once.
Author Response

(The authors gave the same response as above.)
